# Role of the postinspiratory complex in regulating swallow–breathing coordination and other laryngeal behaviors

Alyssa Huff[1], Marlusa Karlen-Amarante[1], Luiz M Oliveira[1], Jan-Marino Ramirez[1,2]*

[1]Center for Integrative Brain Research, Seattle Children's Research Institute, Seattle, United States; [2]Department of Neurological Surgery, University of Washington School of Medicine, Seattle, United States

**Abstract** Breathing needs to be tightly coordinated with upper airway behaviors, such as swallowing. Discoordination leads to aspiration pneumonia, the leading cause of death in neurodegenerative disease. Here, we study the role of the postinspiratory complex (PiCo) in coordinating breathing and swallowing. Using optogenetic approaches in freely breathing anesthetized ChAT-cre:Ai32, Vglut2cre:Ai32 and intersectional recombination of ChATcre:Vglut2FlpO:ChR2 mice reveals PiCo mediates airway protective behaviors. Activation of PiCo during inspiration or the beginning of postinspiration triggers swallow behavior in an all-or-nothing manner, while there is a higher probability for stimulating only laryngeal activation when activated further into expiration. Laryngeal activation is dependent on stimulation duration. Sufficient bilateral PiCo activation is necessary for preserving the physiological swallow motor sequence since activation of only a few PiCo neurons or unilateral activation leads to blurred upper airway behavioral responses. We believe PiCo acts as an interface between the swallow pattern generator and the preBötzinger complex to coordinate swallow and breathing. Investigating PiCo's role in swallow and laryngeal coordination will aid in understanding discoordination with breathing in neurological diseases.

**\*For correspondence:**
jan.ramirez@seattlechildrens.org

**Competing interest:** The authors declare that no competing interests exist.

## Editor's evaluation

This paper is a valuable contribution to the functional characterization of PiCo, a specific medullary region involved in the control of swallow behavior and its coordination with breathing. This study is technically impressive by combining high-skilled approaches to obtain EMG recordings from most of the relevant muscles and nerves involved in swallowing, coupled with elegant intersectional genetic approaches to specifically target for activation cholinergic/glutamatergic neurons that comprise PiCo. The results demonstrate that activation of PiCo is sufficient to trigger swallow events and laryngeal activation in a manner that depends on the timing of the respiratory cycle. Thus the presented data significantly participate in a better understanding of central mechanisms involved in the coordination between swallow and breathing behaviors and in the role of PiCo in postinspiratory behaviors and are critical for the fields of neural control of breathing and swallowing.

## Introduction

The discovery of the preBötzinger complex (preBötC) more than 30 years ago (*Smith et al., 1991*) triggered a wave of mechanistic studies aimed at understanding the neuronal determinants that drive inhalation. By contrast, the mechanistic understanding of exhalation lacks far behind. Indeed, often

called 'passive' expiration, this term suggests that expiration is primarily driven by mechanical recoil forces of the lung and may occur without neuronal control. Far from the truth, expiration is complex involving the neuronal control of multiple muscles, and the exquisite coordinated valving of laryngeal and pharyngeal control with other behaviors such as vocalization, coughing, or swallowing.

Exhalation is traditionally subdivided into different phases – exhalation begins with postinspiration or E1 phase (*Bautista and Dutschmann, 2014*; *Dutschmann and Dick, 2012*), followed by late expiration or the E2 phase (*Richter and Smith, 2014*). But there is also 'active expiration' which is associated with the conditional activation of intercostal and abdominal muscles that are recruited during high metabolic demand (*Abdala et al., 2009*; *Flor et al., 2020*; *Molkov et al., 2011*; *Pisanski and Pagliardini, 2019*).

Postinspiratory complex (PiCo) has been defined by a population of interneurons, located within the intermediate reticular nucleus (IRt) that uniquely co-expresses both glutamate and acetylcholine. PiCo is both sufficient and necessary for generating the postinspiratory phase of breathing. Indeed, it was hypothesized that this complex may be involved in various postinspiratory behaviors such as swallowing and vocalization (*Anderson et al., 2016*). This hypothesis was subsequently confirmed as work in the rat has shown this region serves as a relay within the IRt that integrates postinspiratory motor outputs and other non-respiratory central pattern generators (CPGs) such as swallowing, crying, lapping, and whisking (*Toor et al., 2019*; *Dempsey et al., 2021*; *Hartmann and Brecht, 2020*; *Hülsmann et al., 2019*; *Moore et al., 2013*; *Moore et al., 2014*; *Pitts et al., 2021*; *Wei et al., 2022*). These experiments performed independently by various groups in different animal models suggest that PiCo may serve as a hub, mediating various laryngeal postinspiratory behaviors.

The swallow motor pattern involves coordinated bilateral activation of more than 26 pairs of muscles innervated by 5 cranial nerves to ensure proper food/liquid breakdown (oral phase), safe transfer of the bolus (pharyngeal and esophageal phase) to the stomach, and airway protection (*Matsuo and Palmer, 2008*). Swallow motor pattern moves spatiotemporally in a rostral to caudal direction: starting with the oropharynx and ending with the upper and lower esophageal sphincters (*Doty and Bosma, 1956*; *Pitts and Iceman, 2023*; *Thexton et al., 2007*). While this study focuses primarily on the pharyngeal phase of swallow, we record swallow behavior from muscles of the oropharynx deemed 'submental complex' including mylohyoid, geniohyoid, and digastric muscles innervated by the trigeminal, hypoglossal, and facial nerves, respectively (*Erik Van Lunteren, 1988*). These muscles move the hyoid bone and the larynx superior and anterior, under the tongue (*Pitts and Iceman, 2023*). Muscles of the laryngopharynx include thyrohyoid, posterior cricoarytenoid (PCA), thyroarytenoid, and thyropharyngeus, innervated by the recurrent laryngeal nerve (RLN), a branch of the vagus nerve (*Erik Van Lunteren, 1988*; *Nasri et al., 1997*). When the larynx is elevated, the upper esophageal sphincter opens while the 'laryngeal complex' (PCA; lateral, transverse, and oblique arytenoid, cricothyroid, and thyroarytenoid muscles) closes off the airway by adduction of the thyroarytenoid muscle, and the bolus passes through the pharynx to the esophagus (*Pitts and Iceman, 2023*).

In this study, we aimed to further explore the role of PiCo in the coordination of breathing, swallowing, and laryngeal activation (*Bartlett, 1986*; *Bartlett Jr, D, 1989*; *Dutschmann and Dick, 2012*). Using optogenetic techniques, we performed phase-specific activation of cholinergic and glutamatergic PiCo neurons in ChATcre:Ai32 and Vglut2cre:Ai32 mice, respectively. We also developed an intersectional approach to stimulate ChATcre:Vglut2FlpO:ChR2 neurons in order to specifically activate only glutamatergic neurons that co-express ChAT in PiCo region in a spontaneously breathing anesthetized in vivo preparation. All three optogenetic approaches independently resulted in two airway protective behaviors: swallow and laryngeal activation. We hypothesize that PiCo acts as a hub for airway protective behaviors, aiding in swallow–breathing coordination.

## Results

### Characterization of cholinergic/glutamatergic PiCo neurons

Here, we report for the first time a triple conditioned mouse expressing cre in ChAT cells and FlpO in Vglut2 cells enhanced by a red fluorescent protein (tdTomato, Ai65) inserted into the *Rosa26* locus (N = 4 animals). Using ChAT staining and the ChATcre:Vglut2FlpO:Ai65 mouse, we characterized the distribution of ChAT+ neurons and neurons positive for both ChAT and Vglut2 within PiCo, depicted in a coronal segment and heat map (*Figure 5—figure supplement 1A and B*, N = 4 animals). First,

in the rostrocaudal distribution, we found 403 ± 39 ChAT+-expressing neurons, with the most rostral portion of the denominated PiCo neurons located next to the caudal pole of the facial nucleus, extending from –6.6 to –7.3 mm from Bregma level, reaching caudally to the nucleus ambiguus (NAmb) non-compact portion (Bregma –7.3 mm). We also found PiCo neuron slightly medial to the NAmb extending 500 μm medial and 600 μm dorsal to the NAmb in a 45° angle. Anatomically similar to the previous ChAT staining, the rostrocaudal distribution in the ChATcre:Vglut2FlpO:Ai65 mice showed 242 ± 12 neurons, also represented by a heat map showing the rostrocaudal and medial–lateral distribution (Figure S1C and D).

## Optogenetic stimulation of neurons in PiCo region regulates swallow and laryngeal activation in a phase-specific manner

### Optogenetic stimulation of ChATcre:Ai32 neurons at PiCo

Activation of ChATcre:Ai32 neurons in the PiCo region leads to laryngeal muscle activation or a swallow dependent on the timing within the respiratory cycle. A two-way ANOVA revealed a significant interaction between time and behavior ($p<0.0001$, df = 4, $F$ = 10.87) in ChATcre:Ai32 mice (N = 10). A post hoc Šidák multiple-comparison test revealed swallow is triggered with a significantly higher probability when ChATcre:Ai32 neurons are activated within the first 10% (p=0.02) of the respiratory cycle. However, there is a significantly higher probability laryngeal muscle activation will occur when ChATcre:Ai32 neurons are activated within 70% (p=0.04) to 90% (p=0.005) of the respiratory cycle (*Figure 1A*).

### Optogenetic stimulation of Vglut2:Ai32 neurons at PiCo

A two-way ANOVA revealed a significant interaction between time and behavior ($p<0.0001$, df = 4, $F$ = 16.68) in Vglut2:Ai32 mice (N = 11). The post hoc Šidák multiple-comparison test revealed a significantly higher probability that swallows will be triggered when Vglut2cre:Ai32 neurons are stimulated in PiCo region within the first 10% (p<0.0001) to 30% (p=0.002) of the respiratory cycle, while laryngeal activation will occur with a significantly higher probability when Vglut2cre:Ai32 neurons are activated within 70% (p=0.04) of the respiratory cycle.

### Optogenetic stimulation of ChATcre:Vglut2FlpO:ChR2 neurons at PiCo

To specifically stimulate PiCo neurons, we used double-conditioned mice expressing cre in ChAT cells and FlpO in Vglut2 cells. We then injected the pAAV-hSyn Con/Fon hChR2(H134R)-EYFP vector into PiCo, resulting in expression of channelrhodopsin (ChR2) in neurons that only co-express ChATcre and Vglut2FlpO. From here on we will refer to these neurons as ChATcre:Vglut2FlpO:ChR2. A two-way ANOVA revealed a significant interaction between time and behavior (p<0.0001, df = 4, $F$ = 23.31) in ChATcre:Vglut2FlpO:ChR2 mice (N = 7). The post hoc Šidák multiple-comparison test revealed that there is a significantly higher probability that laryngeal activation will be stimulated when ChATcre:Vglut2FlpO:ChR2 neurons are activated within 70% (p=0.001) and 90% (p=0.01) of the respiratory cycle.

A two-way ANOVA revealed no difference in the probability of triggering a swallow between ChATcre:Ai32, Vglut2cre:Ai32, and ChATcre:Vglut2FlpO:ChR2 mice (*Figure 1Bi*). However, a two-way ANOVA revealed a significant difference in the probability of stimulating laryngeal activation between genetic types (p=0.02, df = 8, $F$ = 2.50). Tukey's multiple-comparison test revealed that the probability of triggering laryngeal activation in Vglut2cre:Ai32 mice compared to ChATcre:Vglut2FlpO:ChR2 mice is significantly lower at 70% (p=0.04) and 90% (p=0.02) of the respiratory cycle (*Figure 1Bii*). Also, the probability of triggering laryngeal activation in Vglut2cre:Ai32 mice compared to ChATcre:Ai32 mice is significantly lower at 90% (p=0.008) of the respiratory cycle (*Figure 1Bii*).

As a control, stimulation of PiCo, across all stimulation durations, in three Ai32$^{+/+}$ mice and four ChATcre:Vglut2FlpO:ChR2 mice where the ChR2 did not transfect ChATcre:Vglut2FlpO, resulted in no response (*Figure 1—figure supplement 1*).

## Respiratory response to optogenetic stimulation of PiCo

When evaluating the phase shift plots, we divided PiCo stimulated responses into either swallow or non-swallow (*Figure 2*). PiCo activation that resulted in either laryngeal activation or in a 'no-motor

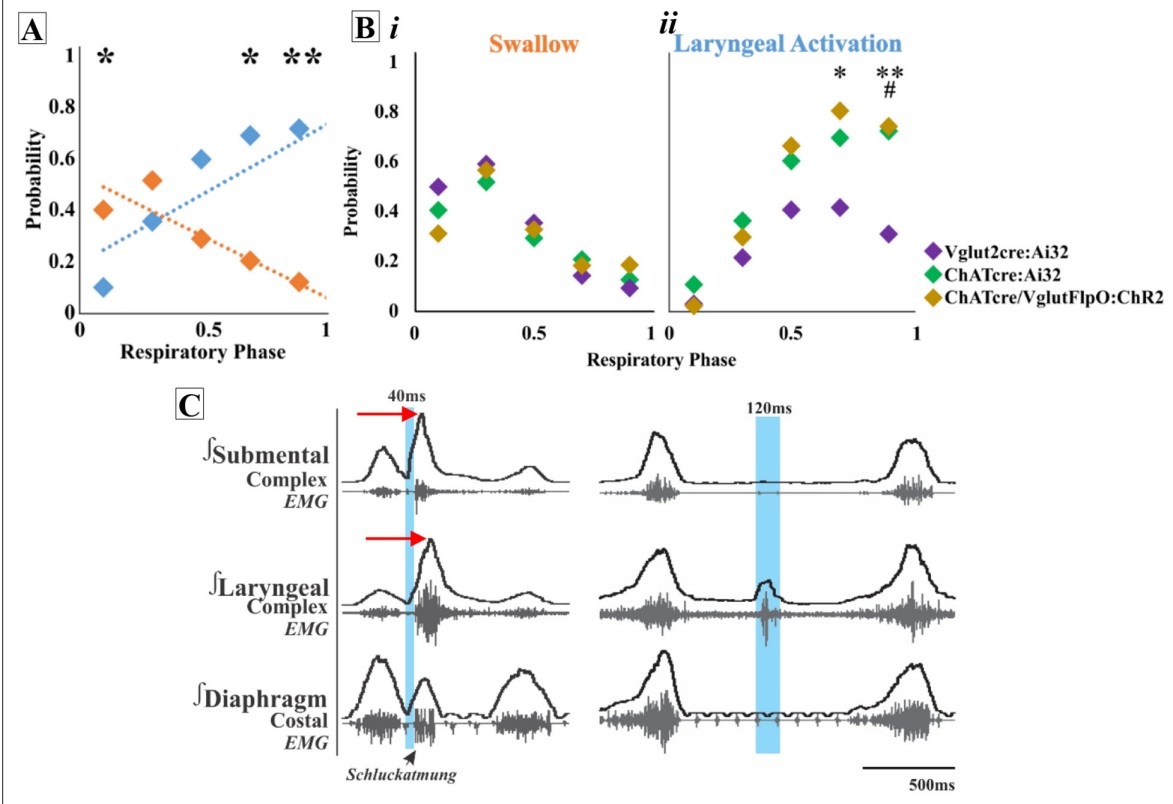

**Figure 1.** Optogenetic stimulation of postinspiratory complex (PiCo) neurons regulates swallow and laryngeal activation in a phase-specific manner. (**A**) Scatter plot of the probability of triggering a swallow (orange) or laryngeal activation (blue) across the respiratory phase (0 start of inspiration, 1 start of next inspiration) in ChATcre:Ai32 mice. * indicates significant increase in the difference between probability of evoking a swallow or laryngeal activation within the first 10% (p=0.02), 70% (p=0.04), and **90% (p=0.005) of the respiratory cycle. (**Bi**) Scatter plot of the probability of triggering a swallow shows no difference between Vglut2cre:Ai32 (purple), ChATcre:Ai32 (green), and ChATcre:Vglut2FlpO:ChR2 (gold) mice. (**Bii**) There is no change in the probability of stimulating laryngeal activation between ChATcre:Ai32 and ChATcre:Vglut2FlpO:ChR2 mice. However, there is a significant difference in the probability between Vglut2cre:Ai32 (purple) and ChATcre:Vglut2FlpO:ChR2 (gold) mice at **70% (p=0.01) and **90% (p=0.003) of the respiratory cycle and Vglut2cre:Ai32 and ChATcre:Ai32 (green) mice at ##90% (p=0.006) of the respiratory cycle. (**C**) Representative traces of PiCo-triggered swallow on the left showing the rostrocaudal swallow motor sequence starting with the peak activation of the submental complex and then the laryngeal complex (red arrows), plus swallow-related diaphragm activation known as *Schluckatmung*. Characterization of laryngeal activation on the right showing only the laryngeal complex is activated in response to the laser in blue.

The online version of this article includes the following figure supplement(s) for figure 1:

**Figure supplement 1.** Stimulation of postinspiratory complex (PiCo) in mice that lack ChR2 or in a region outside of PiCo does not result in activation of PiCo neurons.

**Figure supplement 2.** Prolonged stimulation of postinspiratory complex (PiCo) does not trigger sequential swallow.

response' were considered as non-swallow. We found that in all genetic mouse lines studied, laser pulse duration did not affect the respiratory rhythm reset in either swallow or non-swallow responses, allowing us to group all laser pulse durations as one (*Figure 2—figure supplement 1*). Using a Pearson correlation and simple linear regression, the correlation coefficient (r, *Figure 2A*) and line of best fit (slope, *Figure 2B*), respectively, were calculated for each genetic mouse type and response to determine the degree of correlation between behavior response and reset of the respiratory rhythm. This test reveals that there is a high degree of correlation between shifting or delaying the following inspiratory burst and triggering a swallow when stimulating ChATcre:Ai32 ($r = 0.77$, p<0.0001, slope = 0.87) and ChATcre:Vglut2FlpO:ChR2 ($r = 0.79$, p<0.0001, slope = 0.73) neurons in PiCo region. While there is a moderate degree of correlation in the Vglut2cre:Ai32 ($r = 0.33$, p<0.0001, slope = 0.69) mice. This suggests that triggering a swallow in the ChATcre:Ai32 and ChATcre:Vglut2FlpO:ChR2 mice has a stronger effect on resetting the respiratory rhythm than stimulating glutamatergic neurons in Vglut2cre:Ai32 mice.

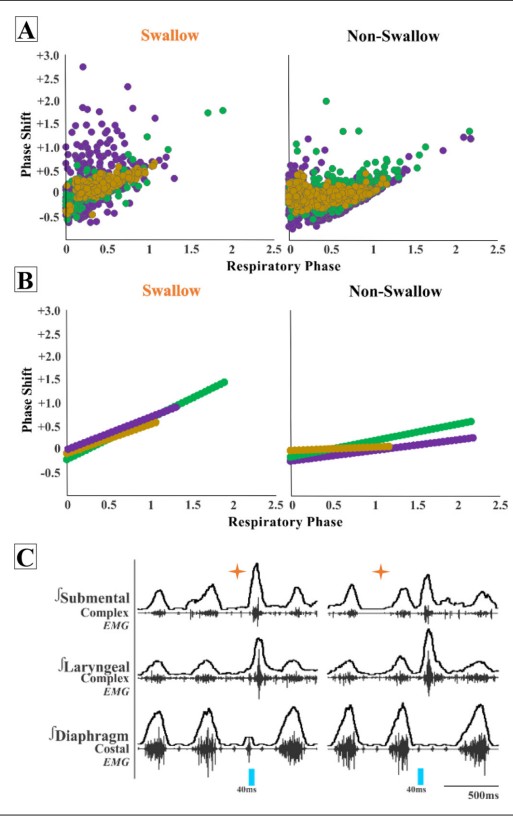

**Figure 2.** Postinspiratory complex (PiCo)-triggered swallows reset the respiratory rhythm, while non-swallows have minimal effect. Respiratory phase shifts plots were divided into two groups: swallow, PiCo activation that triggered a swallow, or non-swallow, PiCo activation that resulted in laryngeal activation or no motor response. (**A**) Individual responses in ChATcre:Vglut2FlpO:ChR2 (gold), ChATcre:Ai32 (green), and Vglut2cre:Ai32 (purple) and (**B**) line of best fit from the above graphs. (**C**) Representative traces of two examples of swallow (orange star) response on respiratory cycle. On the left, PiCo-triggered swallow inhibits inspiration, resulting in an earlier onset of the next inspiratory breath, and on the right a delay in the next inspiration.

The online version of this article includes the following figure supplement(s) for figure 2:

**Figure supplement 1.** Individual responses in ChATcre:Ai32, Vglut2cre:Ai32, and ChATcre:Vglut2FlpO:ChR2 separated by laser pulse duration.

We found a moderate degree of correlation between the following inspiratory burst and non-swallows stimulated in ChATcre:Ai32 ($r = 0.45$, p<0.0001, slope = 0.35) and a low degree of correlation in Vglut2cre:Ai32 ($r = 0.29$, p<0.0001, slope = 0.22) and ChATcre:Vglut2FlpO:ChR2 ($r = 0.23$, p=0.0001, slope = 0.09) mice. This suggests that triggering a swallow has a stronger effect on resetting the respiratory rhythm than activating non-swallows in all of the genetic mouse types under our anesthetic conditions. See 'Discussion' for more considerations.

## Effect of PiCo stimulation duration on swallow behavior and laryngeal activity

We found that regardless of laser pulse duration, ranging from 40 ms to 200 ms, swallows were triggered in an all-or-none manner and had an average duration of 114 ± 19 ms in the ChAT-cre:Vglut2FlpO:ChR2 (*Figure 3*). We conclude that the PiCo-triggered swallow response is not dependent on the duration of the laser pulse. By contrast, PiCo activated laryngeal activity in a graded fashion. As laser pulse duration increased, laryngeal duration increased in an on-off manner where responses to 40 ms pulses were significantly shorter than laryngeal activity stimulated with 200 ms pulses (unpaired *t*-test: p<0.0001, *t* = 11.62, df = 11) (*Figure 3*). This was also true in both the ChATcre:Ai32 (unpaired *t*-test: p=0.02, *t* = 2.90, df = 8) and Vglut2cre:Ai32 (unpaired *t*-test: p=0.004, *t* = 3.68, df = 11) mice (*Supplementary file 1A*).

## Swallow-related characteristics in water-triggered and PiCo-triggered swallows

*Figure 4A* depicts the swallow motor patterns of a water- and PiCo-evoked swallow. A repeated-measures two-way ANOVA revealed no significant differences in swallow onset relative to inspiratory onset between PiCo-evoked and water-evoked swallows (*Figure 4B*). A repeated-measures two-way ANOVA also revealed that PiCo-triggered swallows do not occur at a significantly different time, relative to peak inspiratory diaphragm activity, when compared with water-evoked swallows (*Figure 4C*). All water- and PiCo-triggered swallow-related characteristics in all three genetic mouse lines are reported in *Supplementary file 2*.

PiCo-evoked swallows are characterized by a significant decrease in duration compared to swallows evoked by water in ChATcre:Ai32 (290 ± 125 ms vs. 199 ± 125 ms; paired *t*-test: p=0.02, *t* = 3.05, df = 8), Vglut2cre:Ai32 (256 ± 108 ms vs. 175 ± 94 ms; paired *t*-test: p=0.007, *t* = 3.40, df = 10), and ChATcre:Vglut2FlpO:ChR2 (191 ± 63 ms vs. 114 ± 19 ms; paired *t*-test: p=0.02, *t* = 3.54, df = 5) mice (*Figure 4—figure supplement 1*, *Supplementary file 2*).

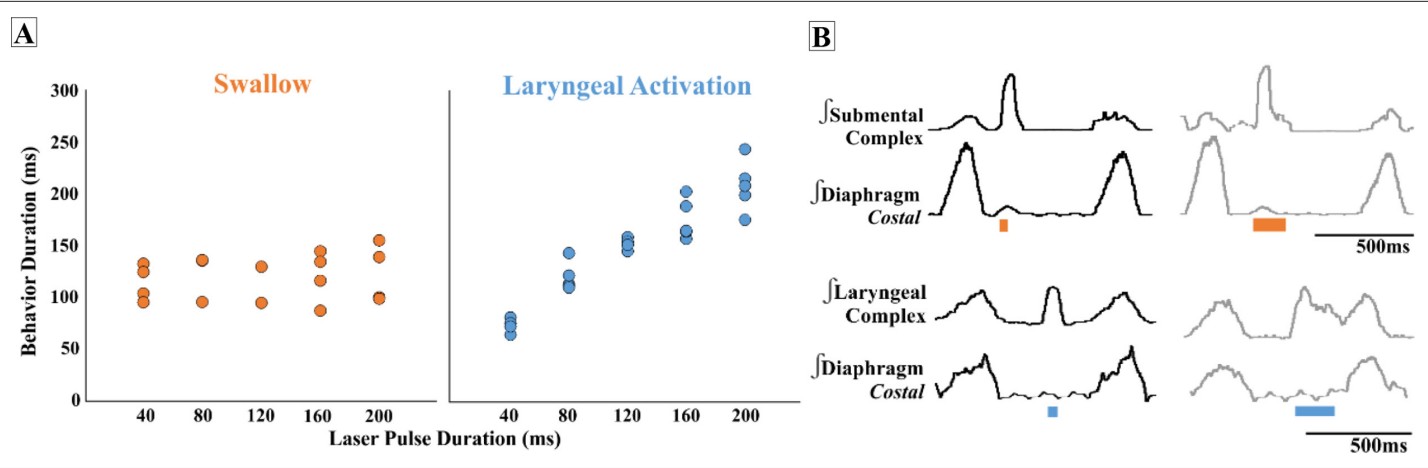

**Figure 3.** Effect of postinspiratory complex (PiCo) stimulation duration on swallow behavior and laryngeal activity. (**A**) Scatter plot of behavior duration versus laser pulse duration for swallow (orange) and laryngeal activation (blue) only in ChATcre:Vglut2FlpO:ChR2 mice. Each dot represents the average duration per mouse. Data for the laryngeal activation analysis, for all genetic mouse lines, is located in *Supplementary file 1A*. (**B**) Representative traces of swallow duration shown by submental complex EMG triggered by 40 ms pulse in orange on the left and 200 ms pulse on the right. Below: representative traces of laryngeal activation, laryngeal complex EMG, duration stimulated by 40 ms pulse in blue on the left, and increases in duration when triggered by 200 ms pulse on the right.

PiCo-evoked swallows have a significant decrease in submental amplitude compared to swallows evoked by water in ChATcre:Ai32 (80 ± 17 vs. 43 ± 40% of max; paired *t*-test: p=0.04, *t* = 2.73, df = 5) and ChATcre:Vglut2FlpO:ChR2 (83 ± 13 vs. 36 ± 33% of max; paired *t*-test: p=0.005, *t* = 4.70, df = 5) mice (*Figure 4—figure supplement 1*, *Supplementary file 2*). Percent of max is measured as a % of the maximum baseline (water swallow) amplitude; see 'Methods'.

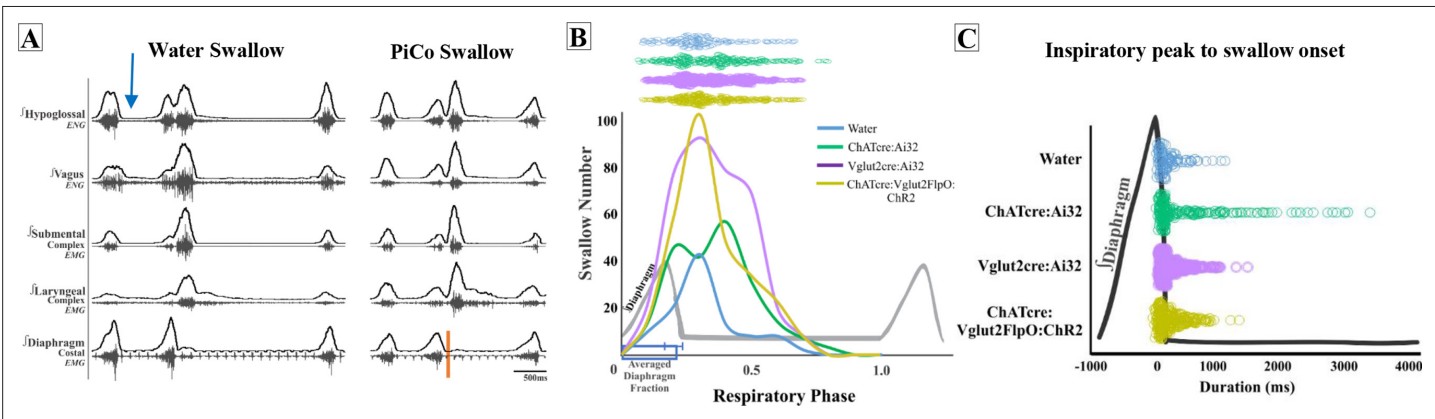

**Figure 4.** Swallow-related characteristics in water-triggered swallows and postinspiratory complex (PiCo)-triggered swallows. (**A**) Representative trace of a swallow triggered by injection of water into the mouth (blue arrow) on the left and PiCo stimulation (orange) on the right. (**B**) Histogram of swallows in relation to the onset of inspiration for water swallows (blue, n = 105), ChATcre:Ai32 (green, n = 214), Vglut2cre:Ai32 (purple, n = 369), and ChATcre:Vglut2FlpO:ChR2 (gold, n = 291). There are more swallows in Vglut2cre:Ai32 mice due to a larger N number and a higher probability of triggering a swallow over any other behavior (*Figure 1B*). (**C**) Dot plot of each swallow in relation to the inspiratory peak. Swallows triggered by water (blue) or PiCo activation occurred at the same time in relation to inspiratory peak. Data for (**B**) and (**C**) are located in *Supplementary file 2*.

The online version of this article includes the following figure supplement(s) for figure 4:

**Figure supplement 1.** Postinspiratory complex (PiCo)-triggered swallows have a decrease in duration and amplitude compared to water-triggered swallows.

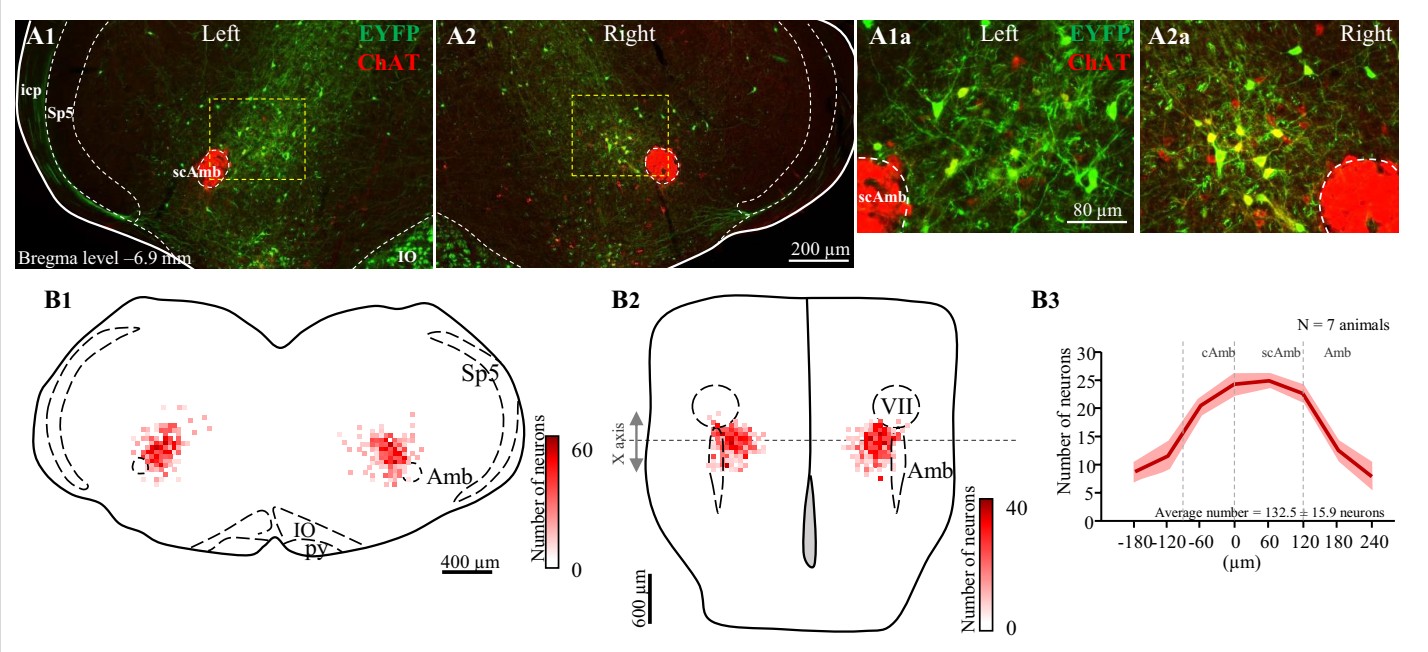

**Figure 5.** Selective transfection of cholinergic/glutamatergic neurons in postinspiratory complex (PiCo) in ChATcre:Vglut2FlpO:ChR2 mice. (**A**) Transverse hemisection through Bregma level (–6.9 mm) of the transfected neurons into PiCo bilaterally, left (**A1**) and right (**A2**), with the pAAV-hSyn Con/Fon hChR2(H134R)-EYFP vector. (**A1a**) magnification of the yellow square in (**A1**) and (**A2a**) magnification of the yellow square in (**A2**). (**B**) Heat map showing the density of neurons transfected by the pAAV-hSyn Con/Fon hChR2(H134R)-EYFP vector from (1) coronal and (2) ventral view of the seven animals used in the functional experiments. X-axis is the transitioning point of compact and semi-compact NAmb. (**B3**) Rostrocaudal distribution of the total number of transfected neurons counted 1:2 series of 25 μm sections into PiCo. cAmb, nucleus ambiguous pars compacta; scAmb, nucleus ambiguus pars semi-compacta; IO, inferior olive; icp, inferior cerebellar peduncle; Sp5, spinal trigeminal nucleus; VII, facial motor nucleus.

The online version of this article includes the following figure supplement(s) for figure 5:

**Figure supplement 1.** Anatomical characterization of postinspiratory complex (PiCo) region.

## Sex-specific differences in swallows triggered by optogenetic stimulation of PiCo region

In ChATcre:Ai32 mice, laryngeal complex duration was significantly shorter in female mice (204 ± 46 ms vs. 109 ± 32 ms; unpaired *t*-test: p=0.03, *F* = 2.05, *t* = 3.01, df = 5) (*Supplementary file 3*). The swallow-related inspiratory delay evoked by stimulating PiCo neurons in Vglut2cre:Ai32 was significantly longer in female mice (273 ± 140 ms vs. 569 ± 256 ms; unpaired *t*-test: p=0.03, *F* = 3.31, *t* = 2.53, df = 9) (*Supplementary file 4*). We were unable to investigate sex-specific differences in ChATcre:Vglut2FlpO:ChR2 mice due to low male and female N numbers.

### Missed or low transfection of PiCo neurons stimulates upper airway motor activity

Post hoc histological analysis was performed in the double-conditioned cre/FlpO mouse to check the transfection of PiCo neurons after injection of the pAAV-hSyn Con/Fon hChR2(H134R)-EYFP vector (ChATcre:Vglut2FlpO:ChR2) (*Figures 5 and 6*). NAmb cholinergic neurons had no transfection, and the rostrocaudal distribution of the transgene-expressing neurons was analyzed as represented in *Figure 5*. Seven ChATcre:Vglut2FlpO:ChR2 mice were stimulated. Swallows and laryngeal activity were triggered in six mice, while in one mouse stimulation evoked only laryngeal activation. In all seven mice, we found that 133 ± 16 neurons expressed EYFP (*Figure 5*). In the six mice where both swallow and laryngeal activation was evoked, we counted an average of 147 transfected neurons while only 46 transfected neurons were counted in the one mouse that only stimulated laryngeal activation.

Interestingly, in four additional mice, activating PiCo resulted in upper airway responses that cannot unambiguously be characterized as either swallows or laryngeal activation as defined before. In these four mice, post hoc histological analysis revealed not only lower total transfection (103 ± 11 neurons),

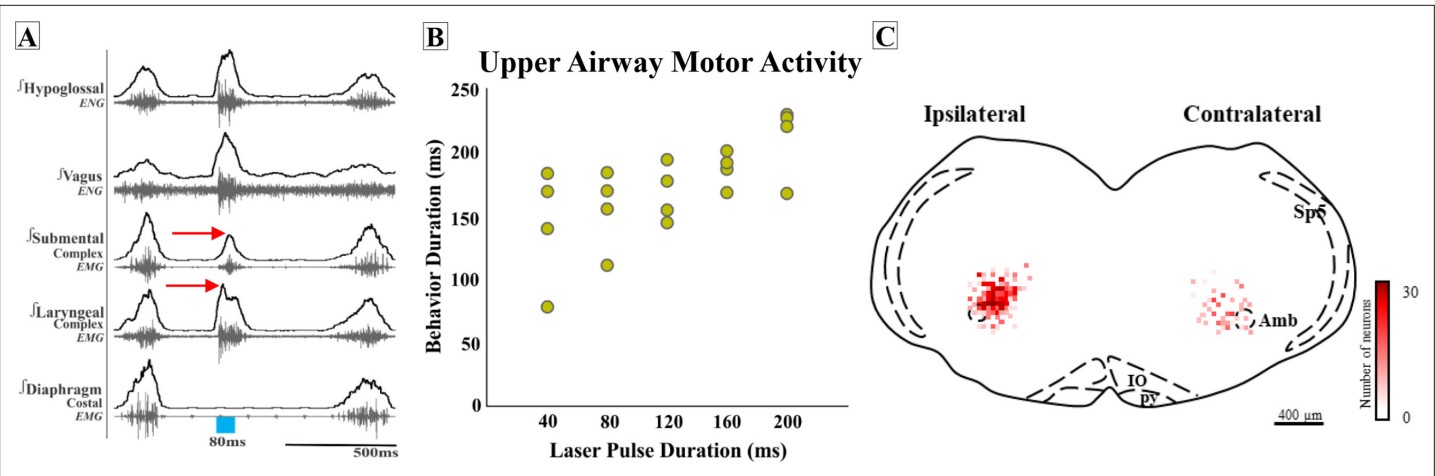

**Figure 6.** Missed or low transfection of postinspiratory complex (PiCo) neurons stimulates upper airway responses that cannot unambiguously be characterized as either swallows or laryngeal activation as defined before. (**A**) Representative trace of 80 ms activation of ChATcre:Vglut2FlpO:ChR2 neurons at PiCo, resulting in an unknown upper airway activation. The red arrows show the laryngeal complex peak activation occurs before the submental complex peak activation; a reverse order from a typical swallow shown in *Figure 1C*. (**B**) Scatter plot of behavior duration versus laser pulse duration for upper airway motor activation. The behavior duration increases as the laser pulse duration increases. Data for this plot is located in *Supplementary file 1B*. (**C**) Heat map showing the density of neurons transfected by the pAAV-hSyn Con/Fon hChR2(H134R)-EYFP vector from coronal view of the four ChATcre:Vglut2FlpO:ChR2 mice. Though bilateral transfection, ipsilateral represents the side of the brainstem with the greatest amount of transfection (69 ± 8 neurons and contralateral 34 ± 4 neurons, N = 4). Amb, nucleus ambiguus; IO, inferior olive; py, pyramidal tract; Sp5, spinal trigeminal nucleus.

but perhaps even more significant, the transfections were asymmetric. We further explored the asymmetric transfection and found the side of the brainstem with the most transfection had an average of 69 ± 8 transfected neurons, and the side with the least amount of transfection had an average of 34 ± 4 transfected neurons (*Figure 6*).

Compared to a typical swallow, the motor sequence in these mice was reversed with laryngeal activity occurring before, instead of after, submental activity (*Figure 6A*). In attempting to identify characteristics of this unusual upper airway motor response, we compared this response to a typical water-triggered swallow. Of note, only two of these four mice swallowed in response to water. The unidentified upper airway motor response had a significant decrease in total behavior duration (236 ± 25 ms vs. 155 ± 14 ms; paired *t*-test: p=0.03, *t* = 20.82, df = 1) (*Supplementary file 5*) much like the comparison between PiCo-triggered swallows and water swallows. We were unable to investigate sex-specific differences in ChATcre:Vglut2FlpO:ChR2 mice due to the low male and female N number.

However, unlike the PiCo-triggered swallow activity, the duration of this upper airway motor activity was dependent on laser pulse duration (*Figure 6B*). The motor activity stimulated by 40 ms pulses was significantly shorter than activity stimulated with 200 ms pulses (paired *t*-test: p=0.01, *t* = 5.61, df = 3) (*Supplementary file 1B*). Thus, this upper airway motor response shares some characteristics of both swallow and laryngeal activation.

## Discussion

In this study, we characterized the neuronal coordination of laryngeal motor activity, swallowing, and breathing by optogenetically stimulating excitatory neurons in PiCo, a region implicated in the control of postinspiratory behaviors (*Anderson et al., 2016*). Our study confirms aspects of Toor et al., who previously hypothesized, under fictive conditions (deafferented and paralyzed), that neurons of the IRt in the PiCo area of rats act as a hub for swallow and laryngeal activity (*Toor et al., 2019*). In this study, we introduced intersectional genetics to specifically stimulate interneurons in the PiCo area that co-express cholinergic and glutamatergic transmitters. This allowed us to compare the effects of these co-expressing neurons with those evoked by stimulating neurons that were only characterized by their cholinergic or glutamatergic transmitter phenotype. We find that activating glutamatergic/cholinergic

interneurons within the PiCo region triggered motor patterns of both swallow and laryngeal-related nerve and muscle activation, specific to respiratory phase activity (*Figure 1*).

## Characteristics of PiCo

Swallow motor activity was evoked in an all-or-none manner, and preferentially when stimulating ChATcre:Vglut2FlpO:ChR2 neurons during inspiration. Swallow motor activity outlasted the triggering stimulus and either abruptly terminated ongoing inspiratory activity or if the stimulus did not terminate inspiration, swallowing occurred after the completion of inspiratory activity. Prolonged stimulation of these neurons (10 s) does not trigger sequential swallows; however, it produces one swallow at the beginning of the stimulus, while inhibiting inspiration, when the stimulus occurs during or right after inspiration (*Figure 1—figure supplement 2*). If prolonged stimulation of PiCo begins further into expiration, either laryngeal activation or no motor response will occur, similar to what is seen with the short pulse stimulations. Thus, the effects of PiCo stimulation are tightly coordinated with the respiratory rhythmogenic network. As already demonstrated by Anderson et al., PiCo is mutually connected with the inspiratory CPG, the preBötC (*Anderson et al., 2016*). Stimulating Dbx1 neurons in the area of the preBötC inhibits ongoing activity in PiCo via GABAergic mechanisms, while stimulating PiCo cholinergic neurons inhibits ongoing inspiratory activity and resets respiratory activity in vitro and in vivo (*Anderson et al., 2016*).

## Postinspiratory airway protective behaviors

Stimulation of PiCo activates different postinspiratory airway protective mechanisms depending on the presence of inspiratory activity. A previous study suggests the presence of a dual peripheral and central mechanism involving vagal lung afferents and central respiratory activity, respectively, dependent on respiratory phase, referred to as a 'swallow gate' (*Horton et al., 2018*). These authors suggest that central processing between areas of the respiratory central pattern generator and the putative swallow pattern generator (SPG) is mediated by a central swallow gate to coordinate swallow and breathing (*Horton et al., 2018*). Here, we show that PiCo functions as a gate. However, for the PiCo-mediated phase-dependent mechanism to occur, bilateral stimulation of a sufficient number of PiCo neurons is necessary since incomplete and/or asymmetric expression of channelrhodopsin in the population of glutamatergic/cholinergic PiCo neurons led to an upper airway motor activity characterized by abnormal temporal activation and coordination between the laryngeal and submental complex (*Figure 6*).

PiCo-stimulated swallow activity was triggered in an all-or-none manner, similar to swallows evoked by liquid, brainstem, and superior laryngeal nerve (SLN) electrical stimulation, a concept established in the late 1800s (*Meltzer, 1883*; *Miller and Sherrington, 1915*). By contrast, laryngeal activation had a graded response dependent on the duration of the optogenetic stimulus (*Figure 3*). As shown by *Anderson et al., 2016*, this laryngeal activity is critical for the generation of postinspiratory activity recorded from the vagal nerve. While PiCo activates laryngeal motor activity during postinspiratory activity, this motor activity could have different functional roles. The laryngeal activity evoked by PiCo stimulation could be a central and integral component to a laryngeal adductor reflex (LAR), an airway protective mechanism that prevents aspiration of foreign material (*Kaneoka et al., 2018*). Another possibility is a muscular response of the, previously studied, non-respiratory expiratory laryngeal motoneurons (ELM). As originally hypothesized by Anderson et al., we believe that PiCo is involved in multiple laryngeal postinspiratory behaviors (*Anderson et al., 2016*) such as vocalization (*Wei et al., 2022*), LAR (laryngeal activation) and coughing.

It has been hypothesized that inhibition of the IRt blocks fictive swallow but not swallow-related apnea. We want to emphasize that this apnea was generated by SLN stimulation and not by a natural swallow stimulation (*Toor et al., 2019*). It is known that SLN stimulation causes endogenous release of adenosine that activates 2A receptors on GABAergic neurons, resulting in the release of GABA on inspiratory neurons and subsequent inspiratory inhibition (*Abu-Shaweesh, 2007*), suggesting that the SLN-evoked apnea may not be the same as a swallow-related apnea. Moreover, microinjections of isoguvacine into the Bötzinger complex attenuated the apneic response but not the ELM burst activity (*Sun et al., 2011*), suggesting that the Bötzinger complex, not PiCo, could be involved in modulating the swallow induced apnea. Future studies will be important to further explore the inhibitory control

of PiCo, in particular connections between the Bötzinger complex and PiCo and their connections to the SPG.

## Interactions between PiCo and other brainstem areas

Kleinfeld and colleagues introduce the concept of pre-premotor regions 'pre²motor' to describe that respiratory oscillators can modulate other orofacial premotor oscillators such as whisking and sniffing (*McElvain et al., 2018*; *Moore et al., 2013*; *Moore et al., 2014*) and recently described swallow (*Huff et al., 2022*; *Pitts et al., 2021*). By modulating the preBötzinger complex, swallow can be shifted to different times of the respiratory cycle as well as changing swallow-related amplitude, laryngeal duration, and motor pattern sequence (*Huff et al., 2022*). Triggering swallow via PiCo activation results in a phase delay of the respiratory cycle, resetting the rhythm, whereas laryngeal activation (non-swallow) has a weaker effect on respiration phase (*Figure 2*). Activation of PiCo-specific neurons arrested or abrogate inspiration triggering swallow, further indicating swallow's hierarchical control over breathing (*Figure 2C*; *Dick et al., 1993*; *Huff et al., 2022*; *Miller and Sherrington, 1915*; *Pitts et al., 2018*).

A previous study in the rat anatomically describes neuronal projections from Kölliker-Fuse and NTS to PiCo (*Oliveira et al., 2021*), suggesting that PiCo's involvement in swallow–breathing coordination also involves the Parabrachial Nucleus and Kölliker-Fuse Nucleus in the dorsolateral pons. These areas have previously been implicated as a sensory relay for the larynx, in particular postinspiratory activity of laryngeal adductor and swallow–breathing coordination (*Bautista and Dutschmann, 2014*; *Dutschmann and Herbert, 2006*). Further studies are necessary to understand the functional interactions between PiCo and the pontine respiratory group on modulating swallow and other airway protective behaviors.

## PiCo and the swallow pattern generator

Behaviorally occurring swallows in response to food or liquid have to be dynamic and alter their timing, duration, and amplitude to accommodate for change in bolus size, texture, and consistency via afferent sensory feedback mechanisms (*Dantas et al., 1990*; *Hrycyshyn and Basmajian, 1972*). There are known projections from NTS to PiCo (*Oliveira et al., 2021*) and connections between PiCo and preBötC (*Anderson et al., 2016*), giving reason to suggest that PiCo acts as an interface between the SPG and preBötC. Here, we show that PiCo-triggered swallows preserve the rostrocaudal swallow pattern and hyoid elevation seen in physiological swallows (*Doty and Bosma, 1956*; *Thexton et al., 2007*), suggesting excitatory inputs between PiCo and the SPG. However, PiCo-triggered swallows occur at a broader range of the respiratory cycle, while still predominantly occurring within the postinspiratory phase (*Figure 4*). We also find that duration and amplitude of swallow-related muscle and nerve activity were decreased during PiCo-stimulated swallows compared to water-evoked swallows (*Figure 4—figure supplement 1*). These differences are likely due to the fact that the PiCo-triggered swallows are missing the behavioral context of water-evoked swallows and do not activate the sensory component of the SPG to the same extent as the water-evoked swallows.

## 'All-or-none' behavior

Swallow has been thought of as an 'all-or-nothing' response as early as 1883 (*Meltzer, 1883*). Whether modulating spinal or vagal feedback (*Huff et al., 2020*), central drive for swallow/breathing (*Huff et al., 2022*), or lesions in swallow related areas of the brainstem (*Jean and Car, 1979*; *Doty et al., 1967*; *Wang and Bieger, 1991*), swallow either occurred or did not. Swallows are thought to be a fixed action pattern, with duration of stimulation having no effect on behavior duration (*Figure 3*; *Dick et al., 1993*). Thus, it was interesting that in instances when few PiCo neurons were transfected, either unilateral or bilateral, an unknown activation of upper airway activity occurred. Motor activity no longer outlasted laser stimulation but rather was contained within, and the timing of the motor sequence was reversed in comparison to a water- or PiCo-evoked swallow (*Figure 6*). Thus, if insufficient numbers of neurons are activated, PiCo's influence to specifically activate swallow or laryngeal activation is blurred, resulting in the uncoordinated activation of muscles involved in both behaviors, which does not occur in an all-or-none manner. Thus, it is possible that PiCo is involved in assembling the swallow motor pattern itself and unilateral activation of PiCo could either desynchronize swallow

interneurons or activate only one side of the SPG. Since we did not record bilateral swallow-related muscles and nerves, this question needs to be further examined.

### *Schluckatmung*

Diaphragm activity has been shown to be multimodal, having different activity patterns for swallow and breathing, including concurrent inhibited respiratory-related activity and activated swallow-related activity, *Schluckatmung*, in physiological and fictive swallows (*Huff et al., 2022*; *Pitts et al., 2021*; *Pitts et al., 2018*). Activation of glutamatergic neurons in the PiCo area resulted in swallow-related diaphragmatic activation, *Schluckatmung* (*Figure 1C*). It has been hypothesized that the SPG activates pre-motor neurons in the dorsal respiratory group responsible for diaphragm recruitment during swallow, and inspiratory neurons of the medial reticular formation have been shown to increase firing frequency during *Schluckatmung* (*Pitts et al., 2021*; *Pitts et al., 2018*).

### Limitations

The depth of anesthesia is a limiting factor of this study. Anesthesia can decrease, and sometimes abolish, laryngeal postinspiratory activity (*Henderson-Smart et al., 1982*; *Sherrey and Megirian, 1974*), and it can also decrease the excitability of swallow reflex (*D'Angelo et al., 2014*). Here, we found that injection of water into the oral cavity of a ChATcre:Ai32 and a ChATcre:Vglut2FlpO:ChR2 mouse did not evoke a swallow, which is likely due to the depth of anesthesia. Another limiting factor of our preparation is the need to open the tracheal window which disrupts the positive subglottic pressure generated during expiration. This leads to lengthening of expiratory duration and also causes irregular breathing (*Henderson-Smart et al., 1982*). Follow-up studies will be necessary to further investigate postinspiratory behaviors and swallow in alert animals.

### Conclusion

We conclude that PiCo aids in regulating laryngeal coordination during swallow and other behaviors. The identification of PiCo as an important region in swallow–breathing coordination will be critical for a better understanding the mechanisms underlying diseases and disorders with prevalent swallow–breathing discoordination. Leigh syndrome, stroke, and Parkinson's disease, as well as obstructive sleep apnea and chronic obstructive pulmonary disease, all have high incidences of aspiration pneumonia (*Armstrong and Mosher, 2011*; *Cvejic and Bardin, 2018*; *Su et al., 2014*; *Tanaka et al., 2022*; *Won et al., 2021*). Aspiration is the result of a discoordination of laryngeal closure during swallow that allows foreign material to enter into the airway instead of the esophagus. Further investigation into PiCo in the context of various breathing and neurological diseases can lead to potential therapeutic targets or decreasing or even eliminating aspiration-related pneumonia in high-risk populations.

## Methods

### Animals

Adult (P54-131, average P75) male and female mice were bred at Seattle Children's Research Institute (SCRI) and used for all experiments. Vglut2-IRES-cre and ChAT-IRES-cre homozygous breeder lines were obtained from Jackson Laboratories (stock numbers 028863 and 031661, respectively). Cre mice were crossed with homozygous mice containing a floxed STOP channelrhodopsin-2 fused to an EYFP (Ai32) reporter sequence from Jackson Laboratories (stock number 024109). Vglut2-IRES-cre crossed with Ai32 will be reported as Vglut2:Ai32 and the ChAT-IRES-cre crossed with Ai32 as ChAT:Ai32. ChAT-IRES-cre and Slc17a6-IRES2-FlpO-D, technically known as 129S-Slc17a6$^{tm1.1(flpo)Hze}$/J, were obtained from Jackson Laboratories (#031661 and #030212, respectively). To generate double-transgenic mice, the ChAT-IRES-cre and Slc17a6-IRES2-FlpO-D strains were interbred to generate compound homozygotes, named as $Chat_{cre}$:$Slc17a6_{FlpO}^{(+/+)}$, which tagged neurons that have a developmental history of expressing both ChAT and Vglut2 and will be reported as ChATcre:Vglut2FlpO. Mice were randomly selected from the resulting litters by the investigators. Offspring were group housed with ad libitum access to food and water in a temperature-controlled (22 ± 1°C) facility with a 12 hr light/dark cycle. All experiments and animal procedures were approved by the Seattle Children's Research Institute's Animal Care and Use Committee (IACUC #0058) and were conducted in accordance with the National Institutes of Health guidelines.

## Brainstem injection of AAV

For the AAV injections, we target the PiCo neurons, first described by *Anderson et al., 2016*, also confirmed by the present results (*Figure 5—figure supplement 1*). We restricted ChR2 expression to the PiCo region in order to transfect and photo-stimulate the region with the highest density of ChATcre:Vglut2FlpO neurons of the PiCo region (*Anderson et al., 2016*). For AVV injection, the mice were anesthetized with isoflurane (2%). The correct plane of anesthesia was assessed by the absence of the corneal and hind-paw withdrawal reflexes. Mice received postoperative ketoprofen (7 mg/kg, subcutaneous [s.c.]) for two consecutive days. All surgical procedures were performed under aseptic conditions. The hair over the skull and neck were removed and skin disinfected. The mice were then placed prone on a stereotaxic apparatus (bite bar set at –3.5 mm for flat skull; David Kopf Instruments Tujunga, CA). A 0.5 mm diameter hole was drilled into the occipital plate on both sides caudal to the parieto-occipital suture. Viral solutions were loaded into a 1.2 mm internal diameter glass pipette broken to a 20 μm tip (external diameter). To target the PiCo region with ChR2-AAV, the pipette was inserted in the brainstem in the following coordinates: 4.8 mm below the dorsal surface of the cerebellum, 1.1 mm lateral to the midline, and 1.6 mm caudal to the lambda, and bilateral injections of 150 nL were made slowly at 50 nL/min using a glass micropipette and an automatic nanoliter injector (NanoinjectII, Drummond Scientific Co. Broomall, PA).

The ChATcre:Vglut2FlpO mice had successful transfection of PiCo neurons by using a pAAV-hSyn Con/Fon hChR2(H134R)-EYFP adenovirus vector (Cat# 55645-AAV8; AddGene, USA; abbreviated as AAV8-ConFon-ChR2-EYFP) herein named ChATcre:Vglut2FlpO:ChR2 in this study. This AAV is a cre-on/FlpO-on ChR2-EYFP under the synapsin promoter and encoded the photoactivatable cation channel channelrhodopsin-2 (ChR2, H134R) fused to EYFP. The vector was diluted to a final titer of 1 × $10^{13}$ viral particles/mL with sterile phosphate-buffered saline.

## In vivo experiments

The same experimental protocol was performed for all Vglut2cre:Ai32, ChATcre:Ai32, ChATcre:Vglut-2FlpO:ChR2 and Ai32$^{+/+}$ mice. Adult mice were initially anesthetized with 100% $O_2$ and 1.5% Isoflurane (Aspen Veterinary Resources Ltd, Liberty, MO) for 2–3 min in an induction chamber. Once the breathing slowed, they were injected with urethane (1.5 mg/kg, i.p. Sigma-Aldrich, St. Louis, MO) and secured supine on a custom surgical table. Core temperature was maintained through a water heating system (PolyScience, Niles, IL) built into the surgical table. Mice were then allowed to spontaneously breathe 100% $O_2$ for the remainder of the surgery and experimental protocol. Adequate depth of anesthesia was determined via heart and breathing rate, as well as lack of toe pinch response every 15 min. A supplemental dose of 0.1 mL of urethane was given to maintain adequate anesthetic depth, when necessary. Bipolar electromyogram (EMG) electrodes were placed in the costal diaphragm to monitor respiratory rate and heart rate throughout the experiment. The trachea was exposed through a midline incision and cannulated caudal to the larynx with a curved (180°) tracheal tube (PTFE 24G, Component Supply, Sparta, TN). The hypoglossal (XII) and vagus (X) nerves were then dissected followed by cannulation of the trachea. The RLN was carefully dissected away from each side of the trachea before the cannula was tied in and sealed with super glue to ensure no damage to the RLN. The trachea and esophagus were then cut to detach the rostral end of the trachea just caudal to the cricoid cartilage, preserving the arytenoids and bilateral RLN. A tube filled with 100% $O_2$ was attached to the cannulated trachea to provide supplemental oxygen throughout the experiment. Continuing in the supine position, the occipital bone was removed, followed by continuous perfusion of the ventral medullary surface with warmed (~36°C) artificial cerebral spinal fluid (aCSF; in mM: 118 NaCl, 3 KCl, 25 NaHCO$_3$, 1 NaH$_2$PO$_4$, 1 MgCl$_2$, 1.5 CaCl$_2$, 30 D-glucose) equilibrated with carbogen (95% $O_2$, 5% $CO_2$) by a peristaltic pump (Dynamax RP-1, Rainin Instrument Co, Emeryville, CA). As previously published (*Figure 6a*; *Huff et al., 2022*), the XII and X nerves were isolated unilaterally, cut distally, and their activity was recorded from a fire-polished pulled borosilicate glass tip (B150-86-15, Sutter Instrument, Novato, CA) filled with aCSF connected to the monopolar suction electrode (A-M Systems, Sequim, WA) and held in a 3D micromanipulator (Narishige, Tokyo, Japan; *Figure 1—figure supplement 1*). Multiple bipolar EMGs, using 0.002″ and 0.003″ coated stainless steel wires (A-M Systems, part nos. 790600 and 79100, respectively), simultaneously recorded activity from several swallow and respiratory-related muscle sites. According to the techniques of *Basmajian and Stecko, 1962*, the electrodes were placed using hypodermic needles 30G (part no. 305106, BD Precision

Glide , Franklin Lakes, NJ) in the *submental complex*, which consists of the geniohyoid, mylohyoid, and digastric muscles, to determine swallow activity. The *laryngeal complex,* consisting of the PCA, lateral, transverse, and oblique arytenoid, cricothyroid, and thyroarytenoid muscles, to determine laryngeal activity during swallow, as well as postinspiratory activity (*Figure 1—figure supplement 1*). The *costal diaphragm*, used to measure the multifunctional activity for both inspiration, as well as *Schluckatmung*, a less common diaphragmatic activation during swallow activity (*Figure 1*). Glass fiber optic (200 um diameter) connected to a blue (447 nm) laser and DPSS driver (Opto Engine LLC, Salt Lake City, UT) was placed bilaterally in light contact with the ventral surface of the brainstem overtop of the predetermined PiCo (*Anderson et al., 2016*; *Figure 1—figure supplement 1*). At the end of the experiment, mice were euthanized by an overdose of anesthetic followed by rapid decapitation or trans-cardial perfusion (see 'Histology section' below).

## Stimulation protocols

(1) Swallow was stimulated by injecting 0.1 cc of water into the mouth using a 1.0 cc syringe connected to a polyethylene tube. (2) 25 pulses of each 40 ms, 80 ms, 120 ms, 160 ms, and 200 ms continuous TTL laser stimulation at PiCo were repeated, at random, throughout the respiratory cycle. (3) 10 s continuous stimulations were repeated three times randomly throughout the respiratory cycle (*Figure 1—figure supplement 2*). The lasers were each set to 0.75 mW and triggered using Spike2 software (Cambridge Electronic Design, Cambridge, UK). These stimulation protocols were performed in all Vglut2cre:Ai32, ChATcre:Ai32, ChATcre:Vglut2FlpO:ChR2 and Ai32$^{+/+}$ mice.

## Analysis

All electroneurogram (ENG) and EMG activity were amplified and band-pass filtered (0.03–1 kHz) by a differential AC Amplifier (A-M Systems model 1700), acquired in an A/D converter (CED 1401; Cambridge Electronic Design), and stored using Spike2 software (Cambridge Electronic Design). Using the Spike2 software, data was further processed using a band-pass filtered (200–700 Hz, 40 Hz transition gap) then rectified and smoothed (20 ms). Using the Spike2 software, the ECGdelete 02.s2s script is used to remove heart artifact, when present, from the ENG and EMG recordings.

We evaluated swallows that were triged by injection of water into the mouth as well as behaviors in response to laser stimulation applied to the PiCo region: swallow, laryngeal activation, and no motor response. Swallow was characterized as a delayed response to the laser outlasting the laser duration, activation of XII, X, submental, and laryngeal complex, and a submental-laryngeal peak activation delay. Diaphragm activity during PiCo-triggered swallows (*Schluckatmung*) was present in some animals but this was not common. Laryngeal activation was characterized as activity of the XII, X, and laryngeal complex from onset to offset of the laser pulse and absence of the diaphragm EMG activity. The submental complex was active in some animals but not all during laryngeal activation. No response was characterized as lack of motor response to the laser and was grouped with laryngeal activation for the non-swallow analysis in respiratory phase shift plots (*Figure 2*). Previously published swallow-related parameters were used to look at swallow–breathing characteristics (*Figure 6*; *Huff et al., 2022*). *Swallow duration* was determined by the onset to the termination of the submental complex EMG activity. In the case the submental complex muscles were not available, then it was determined by the onset to the termination of the XII ENG activity. *Swallow sequence* was calculated as the time difference between the peak activation of the laryngeal and submental complex EMG activity. *Schluckatmung* duration was determined by the onset to the termination of the diaphragm EMG activity during a swallow. *Laryngeal activation duration* was determined by the onset to the termination of the laryngeal complex EMG activity. *Diaphragm inter-burst interval* was calculated as the offset of the diaphragm EMG activity to the onset of the subsequent breath. *Inspiratory delay* was calculated as the offset of the swallow-related laryngeal EMG activity to the onset of the subsequent breath. Duration and amplitude of each nerve and muscle were determined by the onset to the termination of that respective nerve/muscle activity during swallow. All durations are reported in milliseconds (ms), and all amplitudes are reported as a '% of max' calculated as the % of the maximum baseline (water swallow) amplitude.

As previously reported (*Figure 6D*; *Huff et al., 2022*), respiratory phase reset curves were calculated by defining the respiratory cycle as the onset of the diaphragm to the onset of the subsequent diaphragm activity. The *phase shift* elicited by each stimulation of water was calculated as the duration

of the respiratory cycle containing the stimulus, divided by the preceding respiratory cycle. The phase of the swallow stimulation (*respiratory phase*) was calculated as the time between the onset of the inspiration (diaphragm) and the stimulus onset, divided by the expected phase. The average phase shift was then plotted against the respiratory phase in bins containing 1/10 of the expected phase (*Baertsch et al., 2018*). Swallow histogram plots were created by the phase of breathing in which swallow occurred in, calculated as the onset of inspiration to the onset of swallow divided by the respiratory cycle duration and plotted against the number of swallows that occurred within the 1/10 binned respiratory phase (*swallow onset: insp onset*). Swallow was also plotted in relation to the peak activation of the diaphragm as a duration with zero equaling the peak of the inspiratory related diaphragm activity (*swallow onset: insp peak*).

Probability plots were calculated by assigning a '0' to the no response behavior or a '0 or 1' to the laryngeal activation or swallow behavior. These numbers were then averaged and plotted against the *respiratory phase* and binned to 1/10 of the respiratory phase.

All data are expressed as mean ± standard deviation (SD). Statistical analyses were performed using GraphPad Prism 9 (GraphPad Software, Inc, La Jolla). Differences were considered significant at p≤0.05. Investigators were not blinded during analysis. Sample sizes were chosen on the basis of previous studies.

## Histology

At the end of experiments, animals were deeply anesthetized with 5% isoflurane in 100% oxygen and perfused through the ascending aorta with 20 mL of phosphate-buffered saline (PB; pH 7.4) followed by 4% phosphate-buffered (0.1 M; pH 7.4; 20 mL) paraformaldehyde (Electron Microscopy Sciences, Fort Washington, PA). The brains were removed and stored in the perfusion fixative for 4 hr at 4°C, followed by 20% sucrose for 8 hr. A series of coronal sections (25 μm) from the brains were cut using a cryostat and stored in cryoprotectant solution at –20°C (20% glycerol plus 30% ethylene glycol in 50 mL phosphate buffer, pH 7.4) prior to histological processing. All histochemical procedures were done using free-floating sections.

Choline acetyltransferase (ChAT) was detected using a polyclonal goat anti-ChAT antibody (AB144P; Millipore; 1:100), and EYFP was detected using a polyclonal mouse anti-GFP (06-896, Millipore; 1:1000) diluted in PB containing 2% normal donkey serum (017-000-121, Jackson ImmunoResearch Laboratories) and 0.3% Triton X-100 and incubated for 24 hr. Sections were subsequently rinsed in PB and incubated for 2 hr in an Alexa 488 donkey anti-goat antibody (711-545-152; 1:250; Jackson ImmunoResearch Laboratories), or Alexa 647 donkey anti-goat (A31571; 1:400; Life Technologies), or Alexa 488 donkey anti-mouse (715-545-150; 1:250; Jackson ImmunoResearch Laboratories) when appropriated. For all secondary antibodies used, control experiments confirmed that no labeling was observed when primary antibodies were omitted. The sections were mounted on slides in a rostrocaudal sequential order, dried, and covered with fluoromount (00-4958-02; Thermo Fisher). Coverslips were affixed with nail polish.

Sections were also examined to confirm the transfected cells. Section alignment between specimens was done relative to a reference section. The rostral segment of PiCo was identified by the last section with the caudal end of the facial motor neurons and the first section with the rostral portion of the inferior olives. To distinguish PiCo in each section, we used the nucleus ambiguus (Amb), the inferior olives (IO), and the ventral spinocerebellar tract (vsc) as the main anatomic structures. The section that contains the rostral portion of Amb (more densely packed, i.e. cAmb) is the section that contains the rostral portion of PiCo; in a caudal direction, the compacta portion of Amb turns into a semi-compacta portion (scAmb), being aligned as the zero point in the rostral-caudal graphs. Further caudal, the scAmb turns in the non-compacta portion of Amb (*Akins et al., 2017*; *Baertsch et al., 2018*; *Kottick et al., 2017*; *Vann et al., 2018*), characterizing the caudal edge of PiCo. PiCo was also anatomically characterized by immunohistological labeling, revealing ChAT-positive neurons located dorsomedial to c-scAmb and caudal to the facial nucleus as previously described (*Toor et al., 2019*; *Anderson et al., 2016*). As shown in *Figure 5—figure supplement 1*, according to the Paxinos and Franklin mouse atlas (*Kirkcaldie et al., 2012*), the transfected cells were located slightly dorsal to the NAmb near Bregma level –6.84 mm, ~1100 μm from the midline, and ~700 μm above the marginal layer.

## Cell counting, imaging, and data analysis

A VS120-S6-W Virtual Slide Scanner (Olympus) was used to scan all the sections. Images were taken with a color camera (Nikon DS-Fi3). To restrict any influences on our counted results, the photomicrography and counting were performed by one blind researcher. ImageJ (version 1.41; National Institutes of Health, Bethesda, MD) was used for cell counting, and Canvas software (ACD Systems, Victoria, Canada, v. 9.0) was used for line drawings. A one-in-two series of 25 µm brain sections was used per mouse, which means that each section analyzed was 50 µm apart. The area analyzed was delimited based on previously reports (*Anderson et al., 2016*) (mean of 5,423 µm²). The sections were counted bilaterally, averaged, and the numbers reported as mean ± standard error of the mean (SEM). Section alignment were relative to a reference section, as previously described (*Anderson et al., 2016*) and based on Paxinos and Franklin (*Kirkcaldie et al., 2012*).

## Acknowledgements

We are grateful to receive the NIH grants P01 HL14454 and Project 2 (awarded to JMR), R01 HL144801 (awarded to JMR), R01 HL151389 (awarded to JMR), R01 HL126523 (awarded to JMR), F32 HL160102-01 (awarded to AH) for funding this project.

## Additional information

### Funding

| Funder | Grant reference number | Author |
|---|---|---|
| National Institutes of Health | P01 HL14454 and Project 2 | Jan-Marino Ramirez |
| National Institutes of Health | R01 HL144801 | Jan-Marino Ramirez |
| National Institutes of Health | R01 HL151389 | Jan-Marino Ramirez |
| National Institutes of Health | R01 HL126523 | Jan-Marino Ramirez |
| National Institutes of Health | F32 HL160102-01 | Alyssa Huff |

The funders had no role in study design, data collection and interpretation, or the decision to submit the work for publication.

### Author contributions

Alyssa Huff, Conceptualization, Data curation, Formal analysis, Funding acquisition, Methodology, Writing – original draft; Marlusa Karlen-Amarante, Conceptualization, Data curation, Investigation, Writing – review and editing; Luiz M Oliveira, Conceptualization, Data curation, Formal analysis, Writing – review and editing; Jan-Marino Ramirez, Conceptualization, Funding acquisition, Investigation, Methodology, Project administration, Writing – review and editing

### Author ORCIDs

Alyssa Huff http://orcid.org/0000-0003-2817-251X
Jan-Marino Ramirez http://orcid.org/0000-0002-5626-3999

### Ethics

All experiments and animal procedures were approved by the Seattle Children's Research Institute's Animal Care and Use Committee and were conducted in accordance with the National Institutes of Health guidelines.(IACUC #0058).

### Decision letter and Author response

Decision letter https://doi.org/10.7554/eLife.86103.sa1
Author response https://doi.org/10.7554/eLife.86103.sa2

# Additional files

## Supplementary files

• Supplementary file 1. Means and standard deviations (SD) for vagus and laryngeal complex duration during (**A**) laryngeal activation in response to increasing stimuli in ChATcre:Ai32, Vglut2cre:Ai32, and ChATcre:Vglut2FlpO:ChR2 mice and (**B**) upper airway motor activation in response to increasing stimuli in ChATcre:Vglut2FlpO:ChR2 mice.

• Supplementary file 2. Means, standard deviations (SD), p-values, t-statistic (*t*), degrees of freedom (df), from a paired *t*-test, and the direction of change for swallow-related parameters when evoked by water (water swallows) and optogenetic stimulation of PiCo in (**A**) ChATcre:Ai32, (**B**) Vglut2cre:Ai32, and (**C**) ChATcre:Vglut2FlpO:ChR2 mice.

• Supplementary file 3. Means, standard deviations (SD), p-values, F-value, t-statistic (*t*), degrees of freedom (df), from an unpaired *t*-test, and the direction of change for swallow-related parameters between male and female mice during water swallows and PiCo-stimulated swallows in ChATcre:Ai32 mice.

• Supplementary file 4. Means, standard deviations (SD), p-values, F-value, t-statistic (*t*), degrees of freedom (df), from an unpaired *t*-test, and the direction of change for swallow-related parameters between male and female mice during water swallows and PiCo-stimulated swallows in Vglut2cre:Ai32 mice.

• Supplementary file 5. Means, standard deviations (SD), p-values, t-statistic (*t*), degrees of freedom (df), from a paired *t*-test, and the direction of change for swallow-related parameters when evoked by water (water swallows) and optogenetic stimulation of PiCo ChATcre:Vglut2FlpO:ChR2 mice evoking upper airway motor activation.

• MDAR checklist

## Data availability

All data is publicly available https://doi.org/10.6084/m9.figshare.21909819.

The following dataset was generated:

| Author(s) | Year | Dataset title | Dataset URL | Database and Identifier |
|---|---|---|---|---|
| Huff A | 2023 | Huff et al. 2023 Experimental Data Set | https://doi.org/10.6084/m9.figshare.21909819 | figshare, 10.6084/m9.figshare.21909819 |

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
