## [Editor Report]

This paper is a valuable contribution to the functional characterization of PiCo, a specific medullary region involved in the control of swallow behavior and its coordination with breathing. This study is technically impressive by combining high-skilled approaches to obtain EMG recordings from most of the relevant muscles and nerves involved in swallowing, coupled with elegant intersectional genetic approaches to specifically target for activation cholinergic/glutamatergic neurons that comprise PiCo. The results demonstrate that activation of PiCo is sufficient to trigger swallow events and laryngeal activation in a manner that depends on the timing of the respiratory cycle. Thus the presented data significantly participate in a better understanding of central mechanisms involved in the coordination between swallow and breathing behaviors and in the role of PiCo in post-inspiratory behaviors and are critical for the fields of neural control of breathing and swallowing.

---

## [Decision Letter]

**Decision letter after peer review:**

[Editors’ note: the authors submitted for reconsideration following the decision after peer review. What follows is the decision letter after the first round of review.]

Thank you for submitting the paper "Postinspiratory complex acts as a gating mechanism regulating swallow-breathing coordination and other laryngeal behaviors" for consideration by *eLife*. Your article has been reviewed by 3 peer reviewers, and the evaluation has been overseen by a Reviewing Editor and a Senior Editor. The reviewers have opted to remain anonymous.

Comments to the Authors:

We are sorry to say that, after consultation with the reviewers, we have decided that this work will not be considered further for publication by *eLife*, for the following reasons:

Despite the impressive methodological approaches used and some interesting findings provided in the manuscript, the three reviewers reached a consensus considering that the present study suffers from several important weaknesses. First, the conclusions drawn by the authors are not fully and sufficiently supported by the data. Addressing this would require additional sets of experiments.

Second, there is an absolute requirement for much more detail in the description of the methods, more developed and rigorous data analysis including statistics, and a better demonstration of the conceptual framework of phase-dependent PiCo-mediated gating of swallowing and breathing behavior. The concept defended here, meaning that the PiCo should be considered as the swallowing pattern generator remains to be fully demonstrated. With the actual data, the reviewers point to the fact that the authors do not have convincing arguments to affirm this and they rather suggest strongly reconsidering the discussion to address the possibility of other mechanisms possibly involved in the coupling between swallowing and breathing.

Finally, the entire paper would deserve improvement in the writing in general and also in the illustrations to fix some imprecisions, errors, contradictions, and lack of details.

*Reviewer #1 (Recommendations for the authors):*

Abstract:

• I don't think that the current study "reveals this small brainstem microcircuit acts as a central gating mechanism for airway protective behaviors." How is PiCo functioning as a microcircuit in this capacity? The statement is too bold in general and it would be more accurate to say "this region mediates swallow and other laryngeal behaviors".

• Distinguish "laryngeal activation" from swallow more clearly here.

• The sentence beginning "PiCo triggers consistent…" is confusing as written

Introduction:

• The explanation of "yield" as it applies to diaphragm activation needs to be expanded.

• The role of this region with crying is mentioned in the introduction, but there is no citation for papers that discuss the integration of breathing with vocalization. Such citations could include Hartmann and Brecht 2020, Wei et al., 2022, and Hulsmann et al., 2019.

Results:

• The first heading includes "gates" – this is an interpretation, and as such doesn't belong in the results.

• In the results paragraph for Figure 2, the main findings are not stated.

• For the result "laser pulse duration did not affect respiratory rhythm reset…" please include the data/stats for this.

• The headings for Figure 3 results paragraph and for the figure legend don't make sense as written.

• In Figure 4A: please mark where the water stim was applied. This panel is also not explained in the Results section, so please add it there.

• Figure 4B, please add a color legend.

• For Figure 4, please add a table with stats for these data.

• Regarding the swallow numbers in Figure 4, why are there so many more for the Vglut2 strain? Were there more trials or were they more likely to swallow with the stim? It's not clear if absolute swallow numbers are given anywhere – just probabilities.

• In general, the terms used for your transgenic mice/neurons are confusing in places. In the first place you want to use the abbreviations, please define them very clearly.

• Figure S1 and Figure 6: using – and + signs to refer to water vs opto stims is confusing. "-ChAT" or "+ChAT" makes it seem like there's a genetic difference between the two. Pick different X-axis labels to distinguish between stims. The Figure S1 explanation in the results text is also quite confusing.

• Why did you present different metrics for the different strains in Figure S1? LC duration, insp delay, and XII amplitude are done for ChAT but not Vglut2 animals. And the opposite for X burst duration and X amplitude? That makes it difficult to visually compare the different strains.

• Also in Figure S1: I can't clearly see data points/lines for all the animals in all the plots. For example, it doesn't appear that data for all 10 ChAT animals are in each of the purple plots. Just wondering if they were excluded for some reason or if I'm misunderstanding something.

• Figure S2 and Figure 5. Please put directions on the X-axis of the plot and explain your zero point.

• The paragraph (and related figs/tables such as tables S3and7) where you talk about the "different behavior" in 3 mice is confusing, mostly due to the term chosen. Try calling it "partial swallow" or "incomplete swallow" or something like that. A normal swallow is technically "swallow-related motor activation", so it's difficult to distinguish the two with that term.

• In the paragraph that starts with "Swallow related motor activation was not triggered…", the writing in the paragraph should be edited for clarity.

Methods:

• Are you sure you don't mean descending aorta?

• See other comments about explaining genetics and staining more thoroughly.

*Reviewer #2 (Recommendations for the authors):*

The present study investigates the role of the PiCo in swallow-breathing coordination. The study is a follow-up on previously published data that have already shown that the PiCo plays a role in swallow-breathing coordination. The use of selective optogenetic stimulation of subsets of PiCo neurons is laudable. The novelty of the results is unclear. The role of the PiCo in swallow-breathing coordination was already shown and that swallows can reset the respiratory cycle is known for decades. The finding that optogenetic activation of the PiCo during early expiration triggers swallows while PiCo activation during later stages evokes unspecified laryngeal activation seems an interesting result but may need further validation.

Specific Concerns:

Overall the presented data only partially support the conclusions.

"PiCo triggers consistent swallow behavior and preserves physiologic swallow motor sequence"

Such a conclusion would require performing a power spectral analysis of the motor EMGs that compromise a large variety of functional different oral and laryngeal muscle groups to display the physiologic swallow motor sequence. In this context, it should be shown that EMG recordings of the submental complex which covers muscles active during the oral phase of swallowing should precede the onset of the laryngeal complex EMG. In summary, the swallow motor sequence needs to be better analyzed and quantified.

Methods

The description of the methods is short and lacks important details. As written, the only physiologic parameter controlled during experiments was the core temperature. Were blood gas composition, heart rate, and blood pressure also monitored? If yes please provide data. The figures indicate a range of respiratory frequencies (for instance figure 1C stimulation showing swallow is around 80-100 while breathing frequency for the experiment illustrating laryngeal activation seems to 30-40 breaths/mins). The duration of the diaphragm EMG seems to range from around 500ms to 200ms.

Please verify and discuss whether the depth of anesthesia could have significantly affected the outcomes of your study.

Were any criteria for the overall autonomic state of the animal used to determine the start of an experiment?

Please provide the time constant for EMG and nerve signal integration. They seem to be quite variable in the figures.

The approach for the recording of EMG activities and laser placement is unclear. I assume a ventral approach was implemented for nerve and EMG recordings, but it is unclear how you inserted the laser for optogenetic stimulation. The predetermined location of the PiCo is not sufficient information. Apparently, the light activation of the PiCo was performed from the medullary surface (dorsal or ventral??). This seems a crude approach and not well controllable approach. Please provide more detail on the procedure. In addition, since the location of the laser cannot be determined by histology it is mandatory to provide additional controls. Please provide data that show placement at the border of the predetermined location of the Pico does not evoke swallow or laryngeal activation.

Submental and laryngeal complex EMG recordings: How were movement artifacts during swallowing and breathing controlled or avoided? See the comment below – all activity seems synchronized with inspiration in all recordings presented in the results.

Results

Please provide animal numbers for each experimental setting in the text of the result section.

It seems that during breathing both submental and laryngeal complex EMG are synchronized with the diaphragm EMG (see all figures). This would mean that all recorded muscles of the laryngeal and submental complex contract during inspiration. If so, how do the contraction of muscles that are all functionally inspiratory mediate swallowing? Showing a photograph that illustrates the placement of the surface electrodes in the submental and laryngeal complex (please also include labels for all the specific muscles mentioned in the material section) would be helpful to assess the present study and would greatly enhance the impact of the study.

In none of the presented recordings respiratory-related postinspiratory activity is evident in recordings of the laryngeal complex. According to the triple oscillator hypothesis, this would mean that the PiCo, as a postinspiratory oscillator, is inhibited during respiration in the present experimental setting. Could this be an effect of viral transfections?

Strangely, In Figure 6 you show a vagus ENG which has postinspiratory discharge while the laryngeal complex which should include laryngeal adductor muscle activity is quiet. Even more inconsistency is illustrated by the absence of postinspiratory discharge in the vagus in Figure 4. This again raises the question of whether the experimental conditions may affect the outcome of the study.

Figure 6. Has the title " Missed or low transfection of PiCo neurons stimulates incomplete swallow related motor activation" This is also a centerpiece in the discussion.

It is odd that then a representative trace of 80ms activation of ChAT/Vglut2 neurons at PiCo resulting in swallow-related motor activation is shown without highlighting the differences to previous swallow motor activation. The only obvious significant difference was the absence of Schluckathmung. Otherwise, the swallow-related motor activation seems to less or more match the other already quite variable recordings presented, whether Schluckathmung is present or not. Please show a more convincing example of incomplete motor activation, before it is made a central and provocative point in the discussion.

Moreover, Figure 6D illustrates differences between water-evoked swallows and PiCo-evoked swallows which does not relate to the differences between sufficient and insufficient PiCo transfection and belongs rather to Figure 4. – what is the rationale for the organization of this figure?

Figure 4 according to your scale bar the water-evoked swallow is longer in duration and thus does not match the group data illustrated in Figure 6.

Figure 7 is unnecessary, and biased, and does not summarize the findings or the discussion.

Finally, it would also greatly enhance the impact of the study to illustrate whether prolonged PiCo stimulation (1-5s) can evoke multiple swallows – it would be important to know whether the PiCo is only involved in the central gating of solitary swallows or can also gate multiple swallows. This would be also important for the translational impact (as discussed) of your study since patients usually aspirate after performing multiple swallows during drinking behavior.

Tracing experiments:

Where in the brainstem did you actually see CTb-labelled cells after larynx injection? What was the rationale for this experiment and why was it important to study whether the PiCo contains motor neurons? Please better explain why this is a crucial experiment for the proof of concept that the PiCo is involved in swallow-breathing coordination.

CTb NTS: The tracer injection site clearly shows a spillover of CTb into the hypoglossal motor neurons.

Suppl Figure 3D is the pico located in the caudal medulla at the level of the area postrema? Has to be a mistake and thus the figure does not illustrate the absence of CTB-labelled cells in the PiCo.

The discussion needs to be focused on the present results.

For instance, it is not adequate to explain the quite variable burst durations of the putative PiCo-mediated swallow activation by discussing that swallow's pattern may change in relation to the structure and consistency of a food bolus, etc. This was not studied. A discussion that questions the all-or-nothing mechanisms because of incomplete swallow motor activation without showing an example of such incomplete motor activation is also not appropriate. Instead, a focused discussion on why the PiCo evoked motor activations reflect swallows should be included. A discussion of technical limitations (e.g. how anesthesia may have affected the results) is also missing.

*Reviewer #3 (Recommendations for the authors):*

Abstract:

"Co-transmission is confusing in this context, recommend using terminology like "intersectional recombination"

"Viral tracing" when referring to CTB tracing is incorrect. CTB is a molecule (protein complex) not a virus.

Introduction:

The introduction would benefit from a description of the swallowing pattern and the muscles involved, with particular emphasis on those from which EMG recordings were made in the current study.

The 2016 Anderson paper suggests that PiCo is homogeneous (not "heterogenous) with a very specific molecular complement ChAT/Vglut2, a definition to the exclusion of the diverse surrounding neurons.

"Since, work in the rodent has shown this region serves as a premotor relay within the IRt that integrates postinspiratory motor outputs and other non-respiratory central pattern generators (CPGs) such as swallowing, crying, lapping, whisking (Ain Summan Toor et al., 2019; Dempsey et al., 2021; Moore et al., 2013; Moore, Kleinfeld, and Wang, 2014; Pitts, Huff, Reed, Iceman, and Mellen, 2021), suggesting that PiCo may serve as a hub, acting as a gate, for various laryngeal postinspiratory behaviors"

The IRt region has certainly been shown to be premotor the oro-motor behaviours listed. The above claim seems to base the PiCo's potential involvement as a pre-motor structure for laryngeal adduction purely on its proximity to the IRt as opposed to any actual demonstration that PiCo neurons are themselves premotor (i.e. innervate motor neurons). If that is the case, it should be made more explicit in the text, 1. what is being inferred about the PiCo in regard to pre-motor status, and 2. offer some evidence to support that reasoning.

Results:

It is a little laborious to specify the transgenic condition in each experiment (e.g. Vglut-cre – early stimulation vs ChAT-cre – early stimulation vs Vglut2-flpo/ChAT-cre – early stimulation). This study has identified a transgenic solution that permits selective stimulation PiCo neurons (i.e. Cre and flippase recombination of ChR2 in the con/fon AAV system): the legibility of the manuscript might be improved by first establishing equivalence (or inequivalence) between these genetic conditions and then reporting only the results for the intersectional condition that best captures PiCo (and saving the Vglut2- and ChAT- only conditions for the supplementary data). If the Vglut2 "PiCo" stimulation provides a quantitative or qualitatively different response to the double condition, it should probably be reported under the caveat these effects are likely due to the many non-PiCo neurons being stimulated in parallel. Sup Figure 3 B from this study (and Figure 1B from Anderson et al. 2016) demonstrates this quite well, ChAT neurons likely make up less than 1/3 of the vglut2 neurons in the area.

Shortcomings of the anatomical tracing are highlighted in the public review section.

[Editors’ note: further revisions were suggested prior to acceptance, as described below.]

Thank you for resubmitting your work entitled "Postinspiratory complex acts as a gating mechanism regulating swallow-breathing coordination and other laryngeal behaviors" for further consideration by *eLife*. Your revised article has been evaluated by Laura Colgin (Senior Editor) and a Reviewing Editor.

The manuscript has been improved but there are some remaining issues that need to be addressed, as outlined below:

This paper is an extremely valuable contribution in the functional characterization of PiCo, a specific medullary region involved in the control of swallow behavior and its coordination with breathing. This study is technically impressive by combining high-skilled approaches to obtain EMG recordings from most of the relevant muscles and nerves involved in swallowing, coupled with elegant intersectional genetic approaches to specifically target for activation cholinergic/glutamatergic neurons that comprise PiCo. The results demonstrate that activation of PiCo is sufficient to trigger swallow events and laryngeal activation in a manner that depends on the timing of the respiratory cycle. Thus the presented data significantly participate in a better understanding of central mechanisms involved in the coordination between swallow and breathing behaviors and in the role of PiCo in post-inspiratory behaviors and are critical for the fields of neural control of breathing and swallowing.

The authors have properly addressed most of the previous comments. Nevertheless, a few concerns should be considered. The main ones being:

1. Writing: the style is a bit repetitive in the result section, leading to the risk of rendering the main outcomes blurred. One sentence should be added at the end of the abstract as a general conclusion. The introduction should be shortened and more focused. Please add subheadings in the discussion to improve clarity and make it easier to read. Please edit the references.

2. Please clarify whether you think that PiCo might be involved in other post-inspiratory behaviors.

3. Please comment on possible interactions between PiCo, BötC and other pontine respiratory groups.

4. Carefully check the figures and consider the possibility of presenting less supplementary figures that appear redundant for some.

5. Discuss what could be the source of excitatory inputs to PiCo during swallowing under physiological condition as well as the connectivity between this region and the swallowing CPG, a structure not well defined, or other brain regions involved in the reflex as the nTS for example. Even if these points might very probably constitute the future research of the Ramirez group.

Full reviews are detailed below.

*Reviewer #4 (Recommendations for the authors):*

General comments

This is a very interesting study performed by Ramirez group which demonstrated using intersectional genetic approaches in order to selectively stimulate a region in the brainstem already described to be involved in post inspiration, now suggesting to be also involved in swallow/laryngeal motor behaviors. The findings are important and relevant for the field. The manuscript is well written, well illustrated, and the authors already made an extensive revision in the first round when they submitted previously to e*Life*. I have only a few comments and suggestions to clarify some points

1) I totally understand the role of PiCo in the post inspiratory activity, and its more than clear that PiCo has to do with this important phase of the motor behavior, however, post inspiration is not only involved in swallow. The selective activation of PiCo is selective for swallow or other postinspiratory behavior?

2) Pag 13, lines 316-319: "Prolonged stimulation of PiCo (10 seconds) does not trigger sequential swallows, however, produces one swallow at the beginning of the stimulus, inhibiting inspiration, when the stimulus occurs during or right after inspiration (Figure S5)". Does the sentence mean that PiCo can trigger other behaviors?

3) "Microinjections of isoguvacine into the Bötzinger complex attenuated the apneic response but not the ELM burst activity (Sun et al., 2011), suggesting the Bötzinger complex, not PiCo, could be involved in modulating the swallow-induced apnea". I really liked this idea, but will it be important to show a possible interaction between BotC and PiCo? Please comment.

4) The authors comment that further studies are necessary to understand the interaction between PiCo and the pontine respiratory group on modulating swallow and other airway protective behaviors. I agree, but a study in rats using standard neuroanatomy/neurophysiology techniques showed a significant input from KF and PAG regions to PiCo region (PMID = 34582982). Both areas have neurons involved in respiratory behaviours such as swallow, cough and speech.

5) If PiCo evoked swallows are not triggered in a behavioral context, I am wondering when PiCo should be recruited to evoke swallow? What is the physiological condition that the physiology system will need to have PiCo activity?

6) Please correct me if I am wrong. According to the histology and counting described in the Results section we have approximately 60% of neurons in thePiCO region, i.e VGlut/Chat neurons. Am I right?

7) Abstract: I think it will be necessary to add a general conclusion about the data observed in the study.

*Reviewer #5 (Recommendations for the authors):*

In this study, the Authors investigated the role of a medullary region, named Postinspiratory Complex (PiCo), in the mediation of swallow/laryngeal behaviours and their coordination with breathing. This region, first described in 2016, is characterized by the presence of glutamatergic and cholinergic interneurons. Thus, experiments have been performed in single allelic and intersectional allelic recombinase transgenic mice to specifically excite cholinergic/glutamatergic neurons by means of optogenetic approaches, while recording swallow and laryngeal motor response. The data indicate that in anesthetized mice the activation of PiCo neurons during inspiration or at the beginning of the post-inspiratory phase evokes a swallow motor sequence, similar to that induced by water application, while the only laryngeal activation is observed when PiCo neurons are activated in later stages of expiration. Interestingly, the Authors further investigate the anatomical landmarks of PiCo using a complex intersectional transgenic technique to reveal by immunohistochemical analyses the distribution of cholinergic/glutamatergic neurons. They conclude that the PiCo region may be involved in the regulation of laryngeal coordination during swallow and other laryngeal behaviours. These data add in an interesting piece in the ongoing discussion on ponto-medullary structures involved in swallow-breathing coordination.

The Authors' claims and conclusions are mostly justified by their data. However, it should be acknowledged that the impact of the study is to a certain extent limited by the lack of knowledge on the source of excitatory inputs to PiCo during swallowing under physiological condition as well as the connectivity between this region and the swallowing CPG, a structure not well defined, or other brain regions involved in the reflex.

Major strengths of the manuscript:

– The methodological approach is sophisticated and well suited for the experimental question. The in vivo mouse preparation developed for this study takes advantage of a selective optogenetic stimulation of specific cell types with the simultaneous electrophysiological recordings from upper airway muscles involved in respiration and swallowing to assess their motor patterns.

– The choice of the topic. The findings described in this study are interesting since there is growing evidence that this medullary region could to be involved in several functions related to airway protection and other oral behaviours.

– This study fits in with previous works. This work is a logic extension of previous studies from this group on swallowing-breathing coordination with further advances in the technical difficulties of experimental protocols and skills necessary to improve them.

Major weaknesses of the manuscript:

– The writing style is wordy and repetitive in the result section. This makes difficult to follow data presentation. This part needs to be shortened to highlight the main outcomes of the study.

– In my opinion, supplementary material is too redundant (Table S6 and table S7 could be removed since illustrate data from a very low number of animals).

Recommendations for the authors

Introduction:

The Introduction is too long and some efforts should be done to make it shorter and more focussed.

Lines 73-85: It is not necessary to repeat all findings of the previous study by Anderson et al. (2016).

Lines 66-72: please, revise the writing and punctuation and rephrase.

Line 102: remove semicolon.

Line 121: please, remove (Figure 1C). You will cite it in the result section.

Results:

In general, writing should be revised to make it less repetitive.

Line 127: PiCo region

Lines 172-178: Please, revise the punctuation of the sentence (…mice. Indicating that..)

Lines 179-180: "…stimulated by ChATcre:Ai32 (r = 0.45, p< 0.0001, slope = 0.35),…" did you mean "stimulated in ChATcre:Ai32 mice"?

Line 182: again why "…mice. Indicating…?

Lines 206-245: Please, avoid the continuous use of parentheses. E.g. instead of.…evoked by water (290{plus minus} 125ms) (199 {plus minus} 125ms) (paired t-test: p = 0.02, t = 3.05, df = 8), it is better, in my opinion:.…evoked by water (290 {plus minus} 125 ms vs. 199 {plus minus} 125 ms; paired t-test: p = 0.02, t = 3.05, df = 8).

Again e.g.: There is a significant decrease in XII (from 297{plus minus} 129ms to 212 {plus minus} 127ms; paired t-test: p = 0.03, t = 2.65, df = 7) and laryngeal complex duration (from 297 {plus minus} 78ms to 163 {plus minus} 63ms; paired t-test: p = 0.009, t = 4.19, df = 5).

Lines 215-216: "max" of what?

Line 221: Swallows triggered by stimulation of Vglut2cre:Ai32 neurons…

Lines 277-283: The sentences are unclear. Please, rephrase and avoid the use of "ipsilateral" and "contralateral".

Reference list:

The two references Huff et al. 2022a and 2022b are incomplete (volume, pages).

Discussion:

This section is well-written and detailed, but it should be divided into subheadings to improve clarity and make the writing flow smoothly. A section devoted to methodological considerations, including some limitations of the study, should be introduced.

Attention should be paid to the editing of the references (e.g. line 350).

Methods:

Line 494: Please, add the ID number of the approved ethic Protocol. The method of euthanasia needs to be explicitly described.

Line 512: Did you evaluate the spread of the 150 nL injectate?

Line 527: 1.5 mg/kg

Line 531: Please, add the supplemental dose of urethane necessary to maintain anaesthesia.

Line 547:.…the XII and X nerves were isolated unilaterally, cut distally, and their activities were recorded from…

Line 554: delete "and"

Line 557: delete semicolon

Line 564: delete "postinspiratory complex"

Lines 566-572: the entire section should be rewritten to improve clarity (no numbered list).

Line 582:.… in response to laser stimulation applied to the PiCo region:….

Lines 587-588:.…and absence of the diaphragm electromyographic activity.

Line 593:.… to the termination of the submental complex electromyographic activity.

Line 599: did you mean the onset of the subsequent breathing?

Lines 657-659: Please, clarify here how did you chose the reference section.

Figures:

Figure 1. Please, revise the following sentence since it is confounding and there are discrepancies with the plot: (Bii) There is no change in probability of stimulating laryngeal activation between ChATcre:Ai32 and ChATcre:Vglut2FlpO:Ai32 mice, however there is a significant difference between Vglut2cre:Ai32 and * and ChATcre:Vglut2FlpO:Ai32 mice at 70% (p = 0.01) and 90% (p = 0.003) of the respiratory cycle and Vglut2cre:Ai32 and # ChATcre:Ai32 mice at 90% (p = 0.006) of the respiratory cycle.

Figure 5. It is much better to use as reference point the caudal portion of the VII nucleus (see Anderson et al. 2016). So, please change the coordinates accordingly. It is very hard to detect and discriminate the transition point of the compact and semi-compact portions of the nucleus ambiguus, especially with fluorescence images. How did you do that? In addition, it should be useful to see lower magnification sections along with a histological control stained, for example, with cresyl violet to better appreciate the location and anatomical landmarks of the investigated region.

Figure 6: Please, remove ipsilateral and contralateral.

Figure S4: See comments on Figure 5 on the reference point. In addition, if in panel B3 and D3 did you mean the total number of neurons instead of the average number? In panel C5, are you sure that at the level of VII nucleus the nucleus ambiguous (pars compacta) is still present (cfr. Franklin and Paxinos atlas)?

*Reviewer #6 (Recommendations for the authors):*

I have reviewed the previous and current submission, the other Reviewers' comments, and the responses of the corresponding author. My comments are brief as the other Reviewers have noted the caveats, many of which I agree with.

This study is a tour de force combining technically-demanding approaches to obtain EMG recordings from most of the relevant muscles and nerves (submental complex, laryngeal complex, diaphrgam; vagus and hypoglossal nerves), coupled with elegant intersectional genetic approaches to specifically target for activation cholinergic/glutamatergic neurons that comprise PiCo. This strategy avoids nearby glutamatergic neurons in the reticular formation and cholinergic motoneurons in the NA.

The results are clear and striking in that activation of PiCo is sufficient to elicit swallow and laryngeal activation, with the precise timing of each dependent on the phase of the respiratory cycle (swallowing in early PI, laryngeal activation in later I). Moreover, the sequence of events resulting from PiCo activation appear qualitatively similar to those events occuring during a more "physiological" swallow elicited by water administration into the oral cavity. The authors have included the necessary controls, as requested by the Reviewers.

In addition, the authors have further refined the anatomical boundaries of Pico using their intersectional transgenic strategy, showing that Glut/Cholinergic neurons that comprise PiCo are medial to the cholinergic neurons of the NA.

The authors should be congratulated for performing such a rigorous, technically-demanding set of experiments to show a critical function for PiCo in vivo in swallowing, laryngeal activation, and their coordination with inspiration. Their findings clearly show that PiCo is sufficient to elicit swallowing and reset the respiratory rhythm. They have expanded our knowledge regarding PiCo in these functions and have set the stage for future studies that will further delineate the mechanisms by which PiCo drives these behaviors. No study is perfect, this one included. There are obviously caveats that the other Reviewers have raised, but that have been acknowledged by the authors and, in my opinion, have been discussed sufficiently in the paper. An important extension of these findings is to demonstrate that PiCo is necessary for swallowing/larygneal function (and coordination with breathing) in a more natural, physiological setting (i.e. normal swallow of a bolus of water/food etc). What are the ramifications of optogenetic silencing of PiCo on normal swallowing? In addition, as mentioned by other reviewers, the precise neural circuitry through which PiCo acts, and how it is integrated with the nTS and other regions needs to be more clearly defined. It seems as though the authors are working on these questions in a separate study.

In summary, the findings of Huff et al. are a significant advance of our knowledge regarding PiCo and post-inspiratory behaviors. Their study using in vivo recordings from the relevant nerves and muscles, in conjunction with an elegant transectional approach to specifically target PiCo is impressive. The finding that the specific activation of PiCO is sufficient to drive swallowing and laryngeal activation during the post-inspiratory period (and resetting inspiration) is critical for the fields of neural control of breathing and swallowing. They set the stage for future research that expands on the role of PiCo in natural post-inspiratory behaviors, as well as their neurophysiological mechanisms that underlie them.

---

## [Author Response]

[Editors’ note: The authors appealed the original decision. What follows is the authors’ response to the first round of review.]

Comments to the Authors:We are sorry to say that, after consultation with the reviewers, we have decided that this work will not be considered further for publication by eLife, for the following reasons:Despite the impressive methodological approaches used and some interesting findings provided in the manuscript, the three reviewers reached a consensus considering that the present study suffers from several important weaknesses. First, the conclusions drawn by the authors are not fully and sufficiently supported by the data. Addressing this would require additional sets of experiments.

Thank you for acknowledging the impressive methodological advances. As suggested, and to better support our data, we have added new experiments and additional controls. Specifically, we added control experiments such as identical optogenetic protocols that stimulate PiCo in Ai32+/+ mice not crossed with any cre lines, as well as ChATcreVglut2FlpO:ChR2 mice that did not transfect with channelrhodopsin, resulting in no response. This was also shown in the form of an added supplemental figure (Figure S3). We added a supplemental figure showing that long (10s) continuous stimulation of PiCo does not result in sequential swallows, rather only one swallow at the beginning of the stimulation if stimulated during or right after inspiration (Figure S5). We added additional data to prove that stimulation duration did not have an effect on respiratory rhythm allowing us to group all stimulation durations for both “swallow” and “non-swallow” behaviors to evaluate respiration rhythm reset (Figure S1). We added additional ChATcre:Vglut2FlpO:ChR2 mice to increase the N number in order to increase the validity and power of the findings. We also removed tracing experiments of lesser quality (CTB) as well as language surrounding these experiments. The only experiments we did not add was using an opsin or technique that would eliminate or inhibit ChATcre:Vglut2FlpO neurons. While we agree that this would have been a good idea, there is a technical limitation since this technique is not available. Efforts are being made in our lab to make this happen, but we are far from ready to use this, which would delay the publication of this study by more than a year. Indeed, because of the already “impressive methodological advances”, this study is more than 6 years in the making. In the meantime, studies with less sophisticated techniques were published. Thus, we don’t think it is justified to ask for further experiments that are not feasible with the currently available technologies.

Second, there is an absolute requirement for much more detail in the description of the methods, more developed and rigorous data analysis including statistics, and a better demonstration of the conceptual framework of phase-dependent PiCo-mediated gating of swallowing and breathing behavior. The concept defended here, meaning that the PiCo should be considered as the swallowing pattern generator remains to be fully demonstrated. With the actual data, the reviewers point to the fact that the authors do not have convincing arguments to affirm this and they rather suggest strongly reconsidering the discussion to address the possibility of other mechanisms possibly involved in the coupling between swallowing and breathing.

As indicated above we have removed any language claiming PiCo is a swallow gate. Though we would like to point out that we never claimed PiCo should be considered the swallow pattern generator. We do not believe this is the case. We have revised the language to make sure this claim is not suggested. We have also added additional statistical analysis and description of the analysis that we have employed, both in text and supplemental figures. We have also provided a supplemental figure of our methods to aid in clarity of out protocols (Figure S6).

Finally, the entire paper would deserve improvement in the writing in general and also in the illustrations to fix some imprecisions, errors, contradictions, and lack of details.

We have carefully edited, and revised the text and the illustrations, as well as added many figures to be void of any errors, contradictions and lack of details.

Reviewer #1 (Recommendations for the authors):Abstract:• I don't think that the current study "reveals this small brainstem microcircuit acts as a central gating mechanism for airway protective behaviors." How is PiCo functioning as a microcircuit in this capacity? The statement is too bold in general and it would be more accurate to say "this region mediates swallow and other laryngeal behaviors".• Distinguish "laryngeal activation" from swallow more clearly here.• The sentence beginning "PiCo triggers consistent…" is confusing as written

The above points have been incorporated into the revised manuscript. The manuscript now states Activation of PiCo during inspiration or the beginning of postinspiration triggers swallow behavior, while there is a higher probability for stimulating only laryngeal activation (activation of only laryngeal complex) when activated further into expiration, suggesting PiCo’s role in swallow-breathing coordination. PiCo triggered swallow preserves physiologic swallow motor sequence, while stimulated laryngeal activation is dependent on stimulation duration.

Introduction:• The explanation of "yield" as it applies to diaphragm activation needs to be expanded.• The role of this region with crying is mentioned in the introduction, but there is no citation for papers that discuss the integration of breathing with vocalization. Such citations could include Hartmann and Brecht 2020, Wei et al., 2022, and Hulsmann et al., 2019.

The above points have been incorporated into the revised manuscript. The manuscript now states: Postinspiration was originally defined in 1937 by Gesell and White (Gesell and White, 1938) calling it an “after discharge” of the diaphragm, similar to “yield” which refers to the simultaneous muscle activity of the decrementing diaphragm and incrementing expiratory laryngeal muscles, serving as a mechanical brake to slow exhalation (Huff A, 2020a)…. Since, work in the rodent has shown this region serves as a premotor relay within the IRt that integrates postinspiratory motor outputs and other non-respiratory central pattern generators (CPGs) such as swallowing, crying, lapping, whisking (Ain Summan Toor et al., 2019; Dempsey et al., 2021; Hartmann and Brecht, 2020; Hülsmann et al., 2019; Moore et al., 2013; Moore, Kleinfeld, and Wang, 2014; Pitts, Huff, Reed, Iceman, and Mellen, 2021; Wei, Collie, Dempsey, Fortin, and Yackle, 2022), suggesting that PiCo may serve as a hub, mediating various laryngeal postinspiratory behaviors.

Results:• The first heading includes "gates" – this is an interpretation, and as such doesn't belong in the results.

Gate has been removed

• In the results paragraph for Figure 2, the main findings are not stated.

We have added the main findings to figure 2 in the manuscript. The manuscript now states: When evaluating the phase shift plots, we divided PiCo stimulated responses into swallow and non-swallow (Figure 2). Non-swallow included PiCo activation that resulted in both laryngeal activation and no motor response. We found that in all genetic mouse lines studied, laser pulse duration did not affect respiratory rhythm reset in either swallow or non-swallow responses, allowing us to group all laser pulse durations as one (Figure S1). Using a Pearson correlation and simple linear regression, the correlation coefficient (r, Figure 2A) and line of best fit (slope, Figure 2B), respectively, was calculated for each genetic mouse type and response to determine the degree of correlation between behavior response and reset of the respiratory rhythm. There is a high degree of correlation between shifting or delaying the following inspiratory burst and triggering a swallow when stimulating ChATcre:Ai32 (r = 0.77, p< 0.0001, slope = 0.87 ) and ChATcre:Vglut2FlpO:ChR2 (r = 0.79 , p< 0.0001, slope = 0.73) mice. While there is a moderate degree of correlation in the Vglut2cre:Ai32 (r = 0.33, p< 0.0001, slope = 0.69) mice. Indicating that triggering a swallow in the ChATcre:Ai32 and ChATcre:Vglut2FlpO:ChR2 mice has a stronger effect on resetting the respiratory rhythm. We found a moderate degree of correlation between the following inspiratory burst and non-swallows stimulated by ChATcre:Ai32 (r = 0.45, p< 0.0001, slope = 0.35), and a low degree of correlation in Vglut2cre:Ai32 (r = 0.29, p< 0.0001, slope = 0.22), and ChATcre:Vglut2FlpO:ChR2 (r = 0.23, p = 0.0001, slope = 0.09) mice. Indicating that triggering a behavior other than swallow has a weak effect on resetting the respiratory rhythm in all of the genetic mouse types, under our anesthetic conditions, see discussion.

• For the result "laser pulse duration did not affect respiratory rhythm reset…" please include the data/stats for this.

We have added a supplemental figure showing this.

• The headings for Figure 3 results paragraph and for the figure legend don't make sense as written.

We have changed the heading for clarity. It now states: Effect of PiCo stimulation duration on swallow behavior and laryngeal activity

• In Figure 4A: please mark where the water stim was applied. This panel is also not explained in the Results section, so please add it there.

This has been added. The manuscript now says: Figure 4A depicts swallow motor pattern of a swallow evoked by water and a swallow evoked by PiCo stimulation.

• Figure 4B, please add a color legend.

This has been added.

• For Figure 4, please add a table with stats for these data.

The data for figure 4C is located in table S2 as “swallow onset: Insp Peak”

The data for figure 4B has been added to table S2 as “swallow onset: Insp Onset”

• Regarding the swallow numbers in Figure 4, why are there so many more for the Vglut2 strain? Were there more trials or were they more likely to swallow with the stim? It's not clear if absolute swallow numbers are given anywhere – just probabilities.

The same number of trials were done for each mouse strain and animal. However for the Vglut2 mice they were more likely to swallow in response to the laser stimulation. There is also one more Vglut2cre:Ai32 experiment than ChATcre:Ai32, and 5 more Vglut2cre:Ai32 then ChATcre:Vglut2FlpO:ChR2 experiments. Language has been added to the manuscript to make this clear. The manuscript now states: There are more swallows in Vglut2cre:Ai32 mice due to a larger N number and a higher probability of triggering a swallow over any other behavior (Figure 1B).

We indicate total of swallows in response to laser for all 3 mouse strains in the figure 4 legend but do not give the raw values of each swallow per mouse per mouse strain. That data will be publically available after publication.

• In general, the terms used for your transgenic mice/neurons are confusing in places. In the first place you want to use the abbreviations, please define them very clearly.

We have changed the abbreviations to define transgenic mice/neurons more clearly.

• Figure S1 and Figure 6: using – and + signs to refer to water vs opto stims is confusing. "-ChAT" or "+ChAT" makes it seem like there's a genetic difference between the two. Pick different X-axis labels to distinguish between stims. The Figure S1 explanation in the results text is also quite confusing.

Thank you for pointing this out. We have changed the X-axis labels to be clearer. It now states: water (Water stim) and swallows triggered by stimulation of PiCo (PiCo stim).

• Why did you present different metrics for the different strains in Figure S1? LC duration, insp delay, and XII amplitude are done for ChAT but not Vglut2 animals. And the opposite for X burst duration and X amplitude? That makes it difficult to visually compare the different strains.

The point of this figure was not to visually compare across genetic strains. Therefore we only presented significant effects between water and PiCo evoked swallow. However, the complete data are presented in table S2.

• Also in Figure S1: I can't clearly see data points/lines for all the animals in all the plots. For example, it doesn't appear that data for all 10 ChAT animals are in each of the purple plots. Just wondering if they were excluded for some reason or if I'm misunderstanding something.

No data points have been excluded. However some data points are extremely close in value and appear as if they are on top of each other, making it difficult to decipher one point from another. There is also the case as in the ChATcre:Ai32 water stim you do not see 10 connecting points because one of the mice did not show swallows in response to water but did swallow in response to laser stimulation. These instances are reflected in degrees of freedom now added to the data tables and in the text: For example, injection of water into the oral cavity of a ChATcre:Ai32 and a ChATcre:Vglut2FlpO:ChR2 mouse did not evoke a swallow.

• Figure S2 and Figure 5. Please put directions on the X-axis of the plot and explain your zero point.

The X axis has been added to the figures. The zero point corresponds to the transitioning point of compact and semi-compact nucleus ambiguus. This description has been added to the manuscript: *X-axis is the transitioning point of compact and semi-compact NAmb.*

• The paragraph (and related figs/tables such as tables S3and7) where you talk about the "different behavior" in 3 mice is confusing, mostly due to the term chosen. Try calling it "partial swallow" or "incomplete swallow" or something like that. A normal swallow is technically "swallow-related motor activation", so it's difficult to distinguish the two with that term.

We do not feel comfortable calling this a partial or incomplete swallow because we are not sure this is a swallow. Swallow requires activation of related muscles in a rostrocaudal temporal sequence. The behavior that has been evoked here does not have this sequence therefore we cannot call this a swallow or any variation of. We have changed the name to “upper airway motor activity”.

• In the paragraph that starts with "Swallow related motor activation was not triggered…", the writing in the paragraph should be edited for clarity.

We have edited this paragraph to improve clarity. The manuscript now says: However, unlike the PiCo triggered swallow activity, upper airway motor activity stimulated by 40ms pulses were significantly shorter than activity stimulated with 200ms pulses (paired t-test: p = 0.01, t = 5.61, df = 3) (Table S1B). Indicating the upper airway motor activity duration is dependent on laser pulse duration similar to PiCo stimulated laryngeal activation, though shares similar characteristics to both swallow and laryngeal activation.

Methods:• Are you sure you don't mean descending aorta?

Yes we are sure. These perfusion methods have been previously published (Lima, J. C., Oliveira, L. M., Botelho, M. T., Moreira, T. S., and Takakura, A. C. (2018).)

Reviewer #2 (Recommendations for the authors):The present study investigates the role of the PiCo in swallow-breathing coordination. The study is a follow-up on previously published data that have already shown that the PiCo plays a role in swallow-breathing coordination. The use of selective optogenetic stimulation of subsets of PiCo neurons is laudable. The novelty of the results is unclear.

Unfortunately, the reviewer missed the important novelty. No study before has stimulated specifically cholinergic-glutamatergic neurons. This is a fundamental technical advance as it is the first study to show that stimulating specifically this interneuron population has the effects on swallow-breathing coordination. All previous studies most certainly also activated laryngeal motoneurons, which could contribute to a swallow reflex response to the laryngeal motor activation, which would not be caused by stimulating specifically PiCo interneurons.

The role of the PiCo in swallow-breathing coordination was already shown

As stated above, no previous study has stimulated specifically cholinergic-glutamatergic interneurons in PiCo.

…and that swallows can reset the respiratory cycle is known for decades. The finding that optogenetic activation of the PiCo during early expiration triggers swallows while PiCo activation during later stages evokes unspecified laryngeal activation seems an interesting result but may need further validation.Specific Concerns:Overall the presented data only partially support the conclusions."PiCo triggers consistent swallow behavior and preserves physiologic swallow motor sequence"Such a conclusion would require performing a power spectral analysis of the motor EMGs that compromise a large variety of functional different oral and laryngeal muscle groups to display the physiologic swallow motor sequence. In this context, it should be shown that EMG recordings of the submental complex which covers muscles active during the oral phase of swallowing should precede the onset of the laryngeal complex EMG. In summary, the swallow motor sequence needs to be better analyzed and quantified.

With all due respect, the reviewer’s comment seems to be inconsistent. On one hand the reviewer states that it has previously been demonstrated that PiCo coordinates swallowing, yet when we read the Toor study, they have not done what the reviewer seems to require from us. Indeed, the Toor study only used SLN stimulation under fictive conditions (deafferented and paralyzed). Our study also includes EMG recordings, while the Toor study is solely based on nerve recordings. While a power spectrum analysis may be appropriate when using surface electrodes of the submental complex in the human that covers and outputs a sum of the recording from a large variety of various muscle groups, this is not appropriate for animal studies that use wire EMG electrodes in certain muscle groups. Our analysis of swallow motor sequence has previously been published (Huff et al. 2022). Using timing of swallow related EMG activity to evaluate swallow sequence is a widely used and accepted analysis in the field (Pitts et al. 2013, King et al., 2020).

MethodsThe description of the methods is short and lacks important details. As written, the only physiologic parameter controlled during experiments was the core temperature. Were blood gas composition, heart rate, and blood pressure also monitored? If yes please provide data. The figures indicate a range of respiratory frequencies (for instance figure 1C stimulation showing swallow is around 80-100 while breathing frequency for the experiment illustrating laryngeal activation seems to 30-40 breaths/mins). The duration of the diaphragm EMG seems to range from around 500ms to 200ms.

Heart and respiratory rate were monitored and recorded via EMG electrodes. Blood pressure was not able to be measured due to the necessity of an arterial line that this experimental model lacks. To this point arterial blood gases were not measured. We see a range of respiratory rates among all of our mice, this could be due to their internal variability and depth of anesthesia.

Please verify and discuss whether the depth of anesthesia could have significantly affected the outcomes of your study.

The following has been added to the manuscript: Anesthesia can decrease, and sometimes abolish laryngeal postinspiratory activity (Henderson‐Smart, Johnson, and McClelland, 1982; Sherrey and Megirian, 1974). The depth of anesthesia is a limiting factor of this study. For example, injection of water into the oral cavity of a ChATcre:Ai32 and a ChATcre:Vglut2FlpO:ChR2 mouse did not evoke a swallow. This is most likely due to the depth of anesthesia since it is known that anesthesia also decreases the excitability of swallow reflex (D’Angelo et al., 2014). Positive subglottic pressure, generated during expiration, is disrupted during opening of the tracheal window (performed in the proposed anesthetized preparation) leading to lengthening of expiratory duration, and also causing irregular breathing (Henderson‐Smart et al., 1982) another limiting factor of this model. There are no apparent studies that have investigated the neural coordination of postinspiratory behaviors and breathing in the alert mouse. Follow-up studies are desirable to further investigate postinspiratory behaviors and swallowing in alert animals.

However, studying alert animals comes with other experimental limitations. For example, it would be extremely difficult to bilaterally and optogenetically stimulate PiCo with the same specificity as done in the present study, while simultaneously recording from laryngeal muscles and hoping that the mouse is still capable of swallowing naturally. We feel that this is the beyond the scope of the present study.

With all due respect to this reviewer, it seems that this reviewer is not familiar with the technical difficulties associated with performing our experiments in mice. One possible reason why Toor et al. likely performed their study in rats was that rats are much larger, which makes it easier to perform the dissections, monitor blood pressure, etc. However, the downside of using rats is that it does not allow the authors to use the same intersectional genetic approaches that can be done in mice, but not in rats. Thus, it took us much longer to complete the present study. When Toor et al. showed their first results in a SFN poster we were already working on this project for more than one year. To develop the intersectional genetic approach used in the present study took close to four years. It might seem like a minor technical advance to this reviewer and apparently also to reviewer number 1, but we initially worked with the Allen Institute that specifically created a transgenic mouse for us to perform these experiments. But, it took us almost two years of breeding and genotyping, numerous histological controls etc. to finally realize that this method does not work. Hence, we started to develop a viral approach. But, also for the viral approach, we encountered problems. We had several months in which the virus did not express. Thus, we have hundreds of “controls” in which “sham-viral injections” or “transgenic approaches” did not allow us to evoke swallows.

Were any criteria for the overall autonomic state of the animal used to determine the start of an experiment?

In the methods sections we state “Adequate depth of anesthesia was determined via heart and breathing rate, as well as lack of toe pinch response every 15 minutes.”

Please provide the time constant for EMG and nerve signal integration. They seem to be quite variable in the figures.

500ms time bars are present in each figures where a representative EMG/ENG trace is present. Also all values are reported in ms indicated on Y-axis and within the Results section.

We have added to the analysis section: Using the Spike2 software, data was further processed using a band pass filtered (200-700Hz, 40Hz transition gap) then rectified and smoothed (20ms). Using the Spike2 software the ECGdelete 02.s2s script is used to remove heart artifact, when present, from the ENG and EMG recordings.

The approach for the recording of EMG activities and laser placement is unclear. I assume a ventral approach was implemented for nerve and EMG recordings, but it is unclear how you inserted the laser for optogenetic stimulation. The predetermined location of the PiCo is not sufficient information. Apparently, the light activation of the PiCo was performed from the medullary surface (dorsal or ventral??). This seems a crude approach and not well controllable approach. Please provide more detail on the procedure.

We apologize, that we did not make this sufficiently clear. Yes, the laser optical fibers are placed on the surface of the ventral brainstem. The manuscript now states: Continuing in the supine position, the occipital bone was removed, followed by continuous perfusion of the ventral medullary surface with warmed (~36ºC) artificial cerebral spinal fluid … Multiple bipolar EMGs using 0.002” and 0.003” coated stainless steel wires (A-M Systems, Sequim, WA, USA, part no.790600 and 79100 respectively) according to the techniques of Basmajian and Stecko (Basmajian and Stecko, 1962) simultaneously recorded activity from several swallow- and respiratory-related muscle sites and were placed using hypodermic needles 30G …Glass fiber optic (200 μm diameter) connected to a blue (447 nm) laser and DPSS driver (Opto Engine LLC, Salt Lake City, Utah, USA) was placed bilaterally in light contact with the ventral surface of the brainstem overtop of the predetermined postinspiratory complex (PiCo) (Anderson et al., 2016) (Figure S6).

In addition to the citation referring to the previously published EMG and ENG placement methods with figure, we have added supplemental figure 6 to show placement in this manuscript

In addition, since the location of the laser cannot be determined by histology it is mandatory to provide additional controls. Please provide data that show placement at the border of the predetermined location of the Pico does not evoke swallow or laryngeal activation.

This has been added in a supplemental figure 6.

Submental and laryngeal complex EMG recordings: How were movement artifacts during swallowing and breathing controlled or avoided? See the comment below – all activity seems synchronized with inspiration in all recordings presented in the results.

Since these mice are urethane anesthetized, and attached to a surgical table there is no movement artifact. These mice do not move except for muscle contraction related to swallow and/or breathing.

ResultsPlease provide animal numbers for each experimental setting in the text of the result section.

This has been added to the Results section.

It seems that during breathing both submental and laryngeal complex EMG are synchronized with the diaphragm EMG (see all figures). This would mean that all recorded muscles of the laryngeal and submental complex contract during inspiration. If so, how do the contraction of muscles that are all functionally inspiratory mediate swallowing?

It is known that swallow related muscles are multifunctional for breathing. It is possible that this is species specific. Since in cats the use of submental complex is not always active during inspiration. Please note that not all muscles that are active during swallow are active during inspiration (diaphragm), except in the case of Schluckatmung, which has a different motor pattern than its swallow related activity. (see Figure 1)

Showing a photograph that illustrates the placement of the surface electrodes in the submental and laryngeal complex (please also include labels for all the specific muscles mentioned in the material section) would be helpful to assess the present study and would greatly enhance the impact of the study.

We have added supplemental figure 6 illustrating our model. (see above comment)

In none of the presented recordings respiratory-related postinspiratory activity is evident in recordings of the laryngeal complex. According to the triple oscillator hypothesis, this would mean that the PiCo, as a postinspiratory oscillator, is inhibited during respiration in the present experimental setting. Could this be an effect of viral transfections?Strangely, In Figure 6 you show a vagus ENG which has postinspiratory discharge while the laryngeal complex which should include laryngeal adductor muscle activity is quiet. Even more inconsistency is illustrated by the absence of postinspiratory discharge in the vagus in Figure 4. This again raises the question of whether the experimental conditions may affect the outcome of the study.

We have combined a response for these two points into one. While there are many possible explanations to this point, we have included a few below.

As published in Anderson et al., PiCo is not always active and not essential for the respiratory off-switch. Though not depicted in the figures of this manuscript we do see intermittent period of laryngeal postinspiratory EMG activity. This is an experimental characteristic that interests us and we will continue to study why the laryngeal postinspiratory activity is variable among all of our mice throughout the experiment. Though we could add a supplemental figure of the intermittent laryngeal postinspiratory activity, we do not feel that this is the main scope of this paper.

The laryngeal complex consist of both inspiratory as well as postinspiratory/expiratory muscles. While it has been established in the cat and rat that the thyroarytenoid muscles is activated throughout postinspiration, we see less of this in the mouse.

We see many instances in which the vagal postinspiratory activity is present while the laryngeal complex post inspiratory activity is not present. This is likely due to the level of anesthesia (refer to the discussion), the effect of removing the portion of the trachea below the level of the larynx, or the overpowering of inspiratory activity in the EMG channel. While it is possible to sparse out the various inspiratory and expiratory laryngeal muscles in the cat and rat, this is extremely difficult to do in the mouse which is why we are measuring a complex of multiple laryngeal muscles not just the thyroarytenoid which is responsible for the laryngeal postinspiratory activity.

Figure 6. Has the title " Missed or low transfection of PiCo neurons stimulates incomplete swallow related motor activation" This is also a centerpiece in the discussion.It is odd that then a representative trace of 80ms activation of ChAT/Vglut2 neurons at PiCo resulting in swallow-related motor activation is shown without highlighting the differences to previous swallow motor activation. The only obvious significant difference was the absence of Schluckathmung. Otherwise, the swallow-related motor activation seems to less or more match the other already quite variable recordings presented, whether Schluckathmung is present or not. Please show a more convincing example of incomplete motor activation, before it is made a central and provocative point in the discussion.

Thank you for bringing this confusion to our attention. We did not add a representative trace of a swallow evoked by water in this particular figure, because representative traces for water-evoked swallows are already shown in figures 1-4. We have also made additional changes to the figure and figure legend to point out the differences between the two behaviors.

Moreover, Figure 6D illustrates differences between water-evoked swallows and PiCo-evoked swallows which does not relate to the differences between sufficient and insufficient PiCo transfection and belongs rather to Figure 4. – what is the rationale for the organization of this figure?

The purpose of Figure 6 (see above comment) is to show characteristics of the PiCo evoked upper airway motor activity that is not identified as either a swallow or laryngeal activation. This behavior shares some characteristics of swallow and some of laryngeal activation (A) shows an example of this behavior where the motor pattern is reversed from a typical swallow with laryngeal complex peak activation happening before the submental complex begins. We have added arrows to more clearly show this. (B) shows that this upper airway motor activation duration is dependent on laser pulse duration much like what is seen with the laryngeal activation behavior. (C) is to show that this incomplete swallow behavior only occurs when there is insufficient number of transfect ChAtcre/Vglut2FlpO cells.

Figure 4 according to your scale bar the water-evoked swallow is longer in duration and thus does not match the group data illustrated in Figure 6.

The water-evoked swallow featured in Figure 4 is not a part of the group data in Figure 6 rather the data is located in figure S1. Data in Figure 6 only include the 4 mice that have an unknown upper airway motor activity.

Figure 7 is unnecessary, and biased, and does not summarize the findings or the discussion.

This figure has been removed

Finally, it would also greatly enhance the impact of the study to illustrate whether prolonged PiCo stimulation (1-5s) can evoke multiple swallows – it would be important to know whether the PiCo is only involved in the central gating of solitary swallows or can also gate multiple swallows. This would be also important for the translational impact (as discussed) of your study since patients usually aspirate after performing multiple swallows during drinking behavior.

Supplemental figure 5 has been added, as well as language in the manuscript has been added to show that prolonged stimulation of PiCo do not evoke sequential swallows. The manuscript now states: Prolonged stimulation of PiCo (10 seconds) does not trigger sequential swallows, however produces one swallow at the beginning of the stimulus, inhibiting inspiration, when the stimulus occurs during or right after inspiration (Figure S5).

Tracing experiments:Where in the brainstem did you actually see CTb-labelled cells after larynx injection? What was the rationale for this experiment and why was it important to study whether the PiCo contains motor neurons? Please better explain why this is a crucial experiment for the proof of concept that the PiCo is involved in swallow-breathing coordination.

We agree with all the reviewers: The CTb tracing experiments do not add anything new to our findings and present with some critical methodological limitations. We have decided to remove that data from our manuscript.

CTb NTS: The tracer injection site clearly shows a spillover of CTb into the hypoglossal motor neurons.

The CTb data has been removed.

Suppl Figure 3D is the pico located in the caudal medulla at the level of the area postrema? Has to be a mistake and thus the figure does not illustrate the absence of CTB-labelled cells in the PiCo.

The CTb data has been removed.

The discussion needs to be focused on the present results.For instance, it is not adequate to explain the quite variable burst durations of the putative PiCo-mediated swallow activation by discussing that swallow's pattern may change in relation to the structure and consistency of a food bolus, etc. This was not studied.

We believe the reviewer has misunderstood. In the discussion we talk about water evoked swallows having an increase in duration compared to PiCo evoked swallows because the lack of sensory information that is likely processed in the PiCo evoked swallows is affecting the duration of swallow. That swallows involving feedback from sensory stimulation such as bolus size, texture, consistency have longer swallows than swallows evoked by PiCo stimulation. However establishing that both are physiologic swallows.

A discussion that questions the all-or-nothing mechanisms because of incomplete swallow motor activation without showing an example of such incomplete motor activation is also not appropriate. Instead, a focused discussion on why the PiCo evoked motor activations reflect swallows should be included.

The reviewer seems to have missed Figure 6 which shows the example of the upper airway motor activity as well as Figures 4 and S1 that compares water swallows and PiCo stimulated swallow. The discussion also reflects these figures.

A discussion of technical limitations (e.g. how anesthesia may have affected the results) is also missing.

We have added in a discussion on technical limitations: Anesthesia can decrease, and sometimes abolish laryngeal postinspiratory activity (Henderson‐Smart et al., 1982; Sherrey and Megirian, 1974). The depth of anesthesia is a limiting factor of this study. For example, injection of water into the oral cavity of a ChATcre:Ai32 and a ChATcre:Vglut2FlpO:ChR2 mouse did not evoke a swallow. This is most likely due to the depth of anesthesia since it is known anesthesia also decreases the excitability of swallow reflex (D’Angelo et al., 2014). Positive subglottic pressure, generated during expiration, is disrupted during opening of the tracheal window (performed in the proposed anesthetized preparation) leading to lengthening of expiratory duration, and also causing irregular breathing (Henderson‐Smart et al., 1982) another limiting factor of this model. There are no apparent studies that have investigated the neural coordination of postinspiratory behaviors and breathing in the alert mouse, a necessity to this field.

Reviewer #3 (Recommendations for the authors):Abstract:"Co-transmission is confusing in this context, recommend using terminology like "intersectional recombination"

This Change has been made: Using optogenetic approaches in freely breathing- anesthetized ChATcre:Ai32, Vglut2cre:Ai32 and intersectional recombination of ChATcre:Vglut2FlpO:ChR2 mice reveals PiCo mediates airway protective behaviors.

"Viral tracing" when referring to CTB tracing is incorrect. CTB is a molecule (protein complex) not a virus.

We have removed the CTB experiments from this manuscript.

Introduction:The introduction would benefit from a description of the swallowing pattern and the muscles involved, with particular emphasis on those from which EMG recordings were made in the current study.

We have added this to the introduction of the manuscript: The swallow motor pattern involves coordinated bilateral activation of more than 26 pairs of muscles innervated by 5 cranial nerves to ensure proper food/liquid breakdown (oral phase), safe transfer of the bolus (pharyngeal and esophageal phase) to the stomach, and airway protection (Matsuo and Palmer, 2008). Swallow motor pattern moves in a rostral to caudal--starting with the oropharynx and ending with the upper and lower esophageal sphincters-- direction (R. W. Doty and Bosma, 1956; Pitts and Iceman, 2023; Thexton, Crompton, and German, 2007). While this study focuses primarily on the pharyngeal phase of swallow, we record swallow behavior from muscles of the oropharynx deemed “submental complex” including mylohyoid, geniohyoid and digastric muscles; innervated by the trigeminal, hypoglossal and facial nerves, respectively (Erik Van Lunteren, 1988). These muscles move the hyoid bone and the larynx superior and anterior, under the tongue (Pitts and Iceman, 2023). Muscles of the laryngopharynx include thyrohyoid, posterior cricoarytenoid (PCA), thyroarytenoid, and thyropharyngeus; innervated by the recurrent laryngeal nerve (RLN), a branch of the vagus nerve (Erik Van Lunteren, 1988; Nasri et al., 1997). When the larynx is elevated, the upper esophageal sphincter opens while the “laryngeal complex” (PCA; lateral, transverse and oblique arytenoid, cricothyroid and thyroarytenoid muscles) closes off the airway by adduction of the thyroarytenoid muscle, and the bolus passes through the pharynx to the esophagus (Pitts and Iceman, 2023).

The 2016 Anderson paper suggests that PiCo is homogeneous (not "heterogenous) with a very specific molecular complement ChAT/Vglut2, a definition to the exclusion of the diverse surrounding neurons.

Unfortunately, this is a common and unintended misunderstanding. We never wanted to imply that PiCo is homogeneous. We used the cholinergic/glutamatergic neurons as a tool to identify and manipulate this region. Nature has strict space limitations and reviewers asked us to remove anything that was tangential to the main thrust of the paper. Thus, we removed evidence that PiCo contains glutamatergic and inhibitory neurons some of which have pacemaker properties. In the original publication we only state: This provides additional evidence that glutamatergic–cholinergic neurons are important for rhythm generation within the PiCo network. We conclude that the PiCo and preBötC can function as independent oscillators with similar rhythm-generating but distinct modulatory properties. We specifically state “important for rhythm generation” in order to not imply that they are the sole source for rhythm generation. We also compare this network with the preBötC which is generally accepted to be “heterogenous” and state “similar rhythm-generating properties”. We also predicted that the PiCo region will be important for other behaviors: As an excitatory rhythmogenic network, the PiCo may not only be involved in the control of breathing, but might also contribute to the generation of other postinspiratory behaviours such as swallowing and vocalization. Although, we did not perform behavioural assays in this study, various types of postinspiratory burst waveforms were observed in the vagal nerve (Extended Data Figure 10) that were similarly affected by manipulation of the PiCo, supporting the idea that this network might have a broad role in controlling postinspiratory activities. **All of our predictions** were confirmed by Toor et al., as well as Kevin Yackle who identified that the same region is critical for swallowing and vocalization. Of course, we are somewhat disappointed that these authors chose different names for the exact same region. But, we refuse to discredit their important contributions.

"Since, work in the rodent has shown this region serves as a premotor relay within the IRt that integrates postinspiratory motor outputs and other non-respiratory central pattern generators (CPGs) such as swallowing, crying, lapping, whisking (Ain Summan Toor et al., 2019; Dempsey et al., 2021; Moore et al., 2013; Moore, Kleinfeld, and Wang, 2014; Pitts, Huff, Reed, Iceman, and Mellen, 2021), suggesting that PiCo may serve as a hub, acting as a gate, for various laryngeal postinspiratory behaviors"The IRt region has certainly been shown to be premotor the oro-motor behaviours listed. The above claim seems to base the PiCo's potential involvement as a pre-motor structure for laryngeal adduction purely on its proximity to the IRt as opposed to any actual demonstration that PiCo neurons are themselves premotor (i.e. innervate motor neurons). If that is the case, it should be made more explicit in the text, 1. what is being inferred about the PiCo in regard to pre-motor status, and 2. offer some evidence to support that reasoning.

For swallow we do not think PiCo is premotor due to the absence of a graded response. Meaning regardless of the stimulation duration if the laser activates PiCo at the appropriate phase during the respiratory cycle a swallow will be triggered. However it is possible for laryngeal activation to be premotor because of its’ graded response. Meaning as the laser duration increases laryngeal activation increases in duration. As stated above, our ongoing anatomical studies fail to demonstrate direct connections to laryngeal motoneurons, but this study has a different scope and is work in progress. Also, we certainly don’t want to wait another year or two before publishing our studies to avoid another situation like this one: i.e. that everything we publish is already known.

Results:It is a little laborious to specify the transgenic condition in each experiment (e.g. Vglut-cre – early stimulation vs ChAT-cre – early stimulation vs Vglut2-flpo/ChAT-cre – early stimulation). This study has identified a transgenic solution that permits selective stimulation PiCo neurons (i.e. Cre and flippase recombination of ChR2 in the con/fon AAV system): the legibility of the manuscript might be improved by first establishing equivalence (or inequivalence) between these genetic conditions and then reporting only the results for the intersectional condition that best captures PiCo (and saving the Vglut2- and ChAT- only conditions for the supplementary data). If the Vglut2 "PiCo" stimulation provides a quantitative or qualitatively different response to the double condition, it should probably be reported under the caveat these effects are likely due to the many non-PiCo neurons being stimulated in parallel. Sup Figure 3 B from this study (and Figure 1B from Anderson et al. 2016) demonstrates this quite well, ChAT neurons likely make up less than 1/3 of the vglut2 neurons in the area.

We have decided to keep this data in the manuscript because it is possible ChAT neurons that do not co expresses Vglut2, and Vglut2 neurons that do not co-express ChAT are PiCo neurons involved in the modulation of swallow and laryngeal activation. We then used the ChATcre:Vglut2:ChR2 mice as a marker to define PiCo region. As stated above and as we published in our Nature paper, PiCo is a rhythmogenic network like the preBötC. While we used the cholinergic and glutamatergic neurons to manipulate this region, this network does not only contain glutamatergic-cholinergic neurons. Indeed, a network containing only one neurotransmitter phenotype would be highly unusual, and to the best of our knowledge, we don’t know of any network in the mammalian nervous system that consists of only one neurotransmitter phenotype. Hence, it makes only sense to us to compare the effect of VGluT2, ChAT and Vglut/ChAT neurons in this region.

[Editors’ note: what follows is the authors’ response to the second round of review.]

The authors have properly addressed most of the previous comments. Nevertheless, a few concerns should be considered. The main ones being:1. Writing: the style is a bit repetitive in the result section, leading to the risk of rendering the main outcomes blurred. One sentence should be added at the end of the abstract as a general conclusion. The introduction should be shortened and more focused. Please add subheadings in the discussion to improve clarity and make it easier to read. Please edit the references.

Thank you for your comments and great suggestions. We have revised the Results section to be less repetitive. We have added a general conclusion sentence to the abstract. We have revised the introduction and removed parts to make it more focused. We have added subsections and have reorganized the discussion accordingly. We have also edited the references.

2. Please clarify whether you think that PiCo might be involved in other post-inspiratory behaviors.

As we already proposed in our Nature article in 2016, we hypothesize that PiCo is a hub for multiple postinspiratory behaviors, similarly to the preBötC which is thought to be involved in multiple inspiratory related behaviors. Indeed, we believe that PiCo is involved in most laryngeal behaviors. In private discussions with Kevin Yackle and as stated in his Wei et al. 2022 publication they have concluded that the iRO is likely a component of PiCo. Thus, it is reasonable to say that PiCo is likely involved in vocalization, Laryngeal adductor reflex and possibly cough. Since rodents do not have the cough reflex this possibility would have to be studied in a different model. The following sentence has been added to the discussion. “As originally hypothesized by Anderson et al. 2016, we believe that PiCo is involved in multiple other laryngeal postinspiratory behaviors such as vocalization (Wei et al., 2022), LAR (laryngeal activation) and coughing.”

3. Please comment on possible interactions between PiCo, BötC and other pontine respiratory groups.

We have added the following sentences to the manuscript: “Future studies will be important to further explore the inhibitory control of PiCo, in particular connections between the Bötzinger complex and PiCo and their connections to the SPG.”

“A previous study in the rat anatomically describes neuronal projections from Kölliker-Fuse and NTS to PiCo (Oliveira, Takakura, and Moreira, 2021) suggesting that PiCo’s involvement in swallow-breathing coordination also involves the Parabrachial Nucleus and Kölliker-Fuse Nucleus in the dorsolateral pons which has previously been implicated as a sensory relay for the larynx, in particular postinspiratory activity of laryngeal adductor and swallow-breathing coordination (Bautista and Dutschmann, 2014; Dutschmann and Herbert, 2006). Further studies are necessary to understand the functional interaction between PiCo and the pontine respiratory group on modulating swallow and other airway protective behaviors.”

4. Carefully check the figures and consider the possibility of presenting less supplementary figures that appear redundant for some.

We thank the reviewer for this suggestion and have removed supplemental tables 6 and 7. We also combined supplemental figures 3 and 6. Thus, we now have 5 supplemental figures. We feel the remaining figures and tables are important for the manuscript.

5. Discuss what could be the source of excitatory inputs to PiCo during swallowing under physiological condition as well as the connectivity between this region and the swallowing CPG, a structure not well defined, or other brain regions involved in the reflex as the nTS for example. Even if these points might very probably constitute the future research of the Ramirez group.

We thank the reviewer for this comment. We agree this is the next step and have started generating preliminary data for these very questions. We have preliminary data using DREADD techniques to silence NTS and find that when NTS is inactivated PiCo cannot trigger swallows, but is still able to reset the respiratory rhythm. However, all of this work is very preliminary and will require several months to be completed. Thus, we feel it is not appropriate for this manuscript. However the following has been added to the manuscript:” There are known projections from NTS to PiCo (Oliveira et al., 2021) and connections between PiCo and preBötC (Anderson et al., 2016) giving reason to suggest PiCo acts as an interface between the SPG and preBötC.”… “Here we show that PiCo-triggered swallows preserve the rostrocaudal swallow pattern and hyoid elevation seen in physiologic swallows (R. W. Doty and Bosma, 1956; Thexton et al., 2007) suggesting excitatory inputs between PiCo and the SPG.”

Reviewer #4 (Recommendations for the authors):General commentsThis is a very interesting study performed by Ramirez group which demonstrated using intersectional genetic approaches in order to selectively stimulate a region in the brainstem already described to be involved in post inspiration, now suggesting to be also involved in swallow/laryngeal motor behaviors. The findings are important and relevant for the field. The manuscript is well written, well illustrated, and the authors already made an extensive revision in the first round when they submitted previously to eLife. I have only a few comments and suggestions to clarify some points1) I totally understand the role of PiCo in the post inspiratory activity, and its more than clear that PiCo has to do with this important phase of the motor behavior, however, post inspiration is not only involved in swallow. The selective activation of PiCo is selective for swallow or other postinspiratory behavior?

We thank the reviewer for this comment. As stated above, we believe that PiCo is important for many laryngeal postinspiratory behaviors. This manuscript focuses on the two we have so far identified being swallow and laryngeal activation. With this study alone we are not able to functionally identify the “laryngeal activity” behavior. However due to previous literature we know laryngeal activity is airway protective so this “laryngeal activity” could be associated with LAR. We have added the following sentence to the manuscript: “As originally hypothesized by Anderson et al. 2016, we believe that PiCo is involved in multiple other laryngeal postinspiratory behaviors such as vocalization (Wei et al., 2022), LAR (laryngeal activation) and coughing.”

2) Pag 13, lines 316-319: "Prolonged stimulation of PiCo (10 seconds) does not trigger sequential swallows, however, produces one swallow at the beginning of the stimulus, inhibiting inspiration, when the stimulus occurs during or right after inspiration (Figure S5)". Does the sentence mean that PiCo can trigger other behaviors?

Yes, much like the short pulse stimulations, If the 10s stimulus occurs later into expiration either laryngeal activation will occur or no motor response will be evoked. This has been clarified in the manuscript and now states: “Prolonged stimulation of these neurons (10 seconds) does not trigger sequential swallows, however produces one swallow at the beginning of the stimulus, while inhibiting inspiration, when the stimulus occurs during or right after inspiration (Figure S5). If prolonged stimulation of PiCo begins further into expiration either laryngeal activation or no motor response will occur, similar to what is seen with the short pulse stimulations.”

3) "Microinjections of isoguvacine into the Bötzinger complex attenuated the apneic response but not the ELM burst activity (Sun et al., 2011), suggesting the Bötzinger complex, not PiCo, could be involved in modulating the swallow-induced apnea". I really liked this idea, but will it be important to show a possible interaction between BotC and PiCo? Please comment.

Yes, we believe it is necessary to understand the connections between PiCo and Bötzinger complex as well as their connections to the putative swallow pattern generator. We are currently collecting preliminary data, for a grant, looking at the inhibitory (GABAergic and glycinergic) control of PiCo. We have also several ongoing studies to try to better define PiCo and Bötzinger complex using Neuropixel recordings as well as RNAseq. But these are ongoing studies and it will take probably more than a year to complete. Since we believe PiCo may play a role in LAR and we know that the Bötzinger complex plays a role in ELM it is necessary to develop an approach that just focuses on investigating the “laryngeal activity” triggered by PiCo. We have added the following sentence to the manuscript: “Future studies will be important to further explore the inhibitory control of PiCo, in particular connections between the Bötzinger complex and PiCo and their connections to the SPG.”

4) The authors comment that further studies are necessary to understand the interaction between PiCo and the pontine respiratory group on modulating swallow and other airway protective behaviors. I agree, but a study in rats using standard neuroanatomy/neurophysiology techniques showed a significant input from KF and PAG regions to PiCo region (PMID = 34582982). Both areas have neurons involved in respiratory behaviours such as swallow, cough and speech.

Thank you for pointing out the missed important publication and known data. We are very embarrassed and cannot believe we did not include this as it is from one of our own authors. The manuscript now states “A previous study in the rat anatomically describes neuronal projections from Kölliker-Fuse and NTS to PiCo (Oliveira, Takakura, and Moreira, 2021) suggesting that PiCo’s involvement in swallow-breathing coordination also involves the Parabrachial Nucleus and Kölliker-Fuse Nucleus in the dorsolateral pons which has previously been implicated as a sensory relay for the larynx, in particular postinspiratory activity of laryngeal adductor and swallow-breathing coordination (Bautista and Dutschmann, 2014; Dutschmann and Herbert, 2006). Further studies are necessary to understand the functional interaction between PiCo and the pontine respiratory group on modulating swallow and other airway protective behaviors.”

5) If PiCo evoked swallows are not triggered in a behavioral context, I am wondering when PiCo should be recruited to evoke swallow? What is the physiological condition that the physiology system will need to have PiCo activity?

Based on very recent preliminary data using DREADD receptor technology, we do not believe that PiCo itself is able to evoke a swallow. It must have input from the caudal NTS. But, to confirm this preliminary finding requires a careful quantitative analysis, careful mapping of the NTS and vector expression. Thus, we believe that PiCo is responsible for the proper timing of swallow. Therefore it is activated in concert with the swallow pattern generator to make sure swallow and inspiration do not occur at the same time. Other ongoing preliminary data in a disease mouse model for Leigh Syndrome shows that in the presence of neuroinflammation in PiCo swallow-breathing coordination is disrupted with swallows occurring during inspiration and not postinspiration. When the neuroinflammation is removed with treatment, swallow-breathing coordination is restored and swallows occur during postinspiration. Since none of this data is published and completed we are not able to reference this in this manuscript. The following has been added in the manuscript: “These differences are likely due to the fact that the PiCo-triggered swallows are missing the behavioral context of water-evoked swallows and do not activate the sensory component of the SPG to the same extent as the water-evoked swallows.

6) Please correct me if I am wrong. According to the histology and counting described in the Results section we have approximately 60% of neurons in thePiCO region, i.e VGlut/Chat neurons. Am I right?

Here we showed 2 different scenarios: (A) ChAT antibody staining representing all ChAT expressing cells in the area; (B) An intersectional genetic animal to tag Chat/Vglut2 double expressing cells. It is possible that other methods will give higher cell counts for cells in these regions than the ChatCre/Vglut2FlpO/Ai65 cross because two recombination events need to take place to activate the reporter. So, for instance, if Cre and Flp were both 80% efficient, the overall efficiency would be only 0.8 x 0.8 = 64%. This contrasts with the use of staining. It’s a general handicap of combinatorial marking systems that multiple events are needed, and as a consequence some sensitivity is sacrificed for the increased specificity.

7) Abstract: I think it will be necessary to add a general conclusion about the data observed in the study.

The following sentence has been added to the abstract: “We believe PiCo acts as an interface between the swallow pattern generator and the preBötzinger complex to coordinate swallow and breathing. Investigating PiCo’s role in swallow and laryngeal coordination will aid in understanding discoordination with breathing in neurological diseases.”

Reviewer #5 (Recommendations for the authors):In this study, the Authors investigated the role of a medullary region, named Postinspiratory Complex (PiCo), in the mediation of swallow/laryngeal behaviours and their coordination with breathing. This region, first described in 2016, is characterized by the presence of glutamatergic and cholinergic interneurons. Thus, experiments have been performed in single allelic and intersectional allelic recombinase transgenic mice to specifically excite cholinergic/glutamatergic neurons by means of optogenetic approaches, while recording swallow and laryngeal motor response. The data indicate that in anesthetized mice the activation of PiCo neurons during inspiration or at the beginning of the post-inspiratory phase evokes a swallow motor sequence, similar to that induced by water application, while the only laryngeal activation is observed when PiCo neurons are activated in later stages of expiration. Interestingly, the Authors further investigate the anatomical landmarks of PiCo using a complex intersectional transgenic technique to reveal by immunohistochemical analyses the distribution of cholinergic/glutamatergic neurons. They conclude that the PiCo region may be involved in the regulation of laryngeal coordination during swallow and other laryngeal behaviours. These data add in an interesting piece in the ongoing discussion on ponto-medullary structures involved in swallow-breathing coordination.The Authors' claims and conclusions are mostly justified by their data. However, it should be acknowledged that the impact of the study is to a certain extent limited by the lack of knowledge on the source of excitatory inputs to PiCo during swallowing under physiological condition as well as the connectivity between this region and the swallowing CPG, a structure not well defined, or other brain regions involved in the reflex.

We agree that it is necessary to understand the excitatory inputs to PiCo. We have started answering this question and preliminary data to suggest PiCo evoked swallows is dependent on caudal NTS, the presumed swallow pattern generator. However, the swallow pattern generator is still not well defined, and further studies will be necessary to better characterize the neuronal substrate of this CPG.

Major strengths of the manuscript:– The methodological approach is sophisticated and well suited for the experimental question. The in vivo mouse preparation developed for this study takes advantage of a selective optogenetic stimulation of specific cell types with the simultaneous electrophysiological recordings from upper airway muscles involved in respiration and swallowing to assess their motor patterns.– The choice of the topic. The findings described in this study are interesting since there is growing evidence that this medullary region could to be involved in several functions related to airway protection and other oral behaviours.– This study fits in with previous works. This work is a logic extension of previous studies from this group on swallowing-breathing coordination with further advances in the technical difficulties of experimental protocols and skills necessary to improve them.

Thank you for these positive comments.

Major weaknesses of the manuscript:– The writing style is wordy and repetitive in the result section. This makes difficult to follow data presentation. This part needs to be shortened to highlight the main outcomes of the study.

Thank you for your comment. We have revised the Results section by reorganizing the subheadings and removing data not pertaining to the overall message of this manuscript. Please see below for a more detailed response.

– In my opinion, supplementary material is too redundant (Table S6 and table S7 could be removed since illustrate data from a very low number of animals).

Table S6 and S7 have been removed.

Recommendations for the authorsIntroduction:The Introduction is too long and some efforts should be done to make it shorter and more focussed.

We have revised the introduction and have removed parts to make it more focused.

Lines 73-85: It is not necessary to repeat all findings of the previous study by Anderson et al. (2016).

We fully agree, but in light of the prior reviews we felt that this was necessary. We now revised this language and removed the detail summary of the Anderson et al. paper. It now states: “PiCo has been defined by a population of interneurons, located within the intermediate reticular nucleus (IRt) that uniquely co-expresses both glutamate and acetylcholine. PiCo is both sufficient and necessary for generating the postinspiratory phase of breathing. Indeed, it was hypothesized that this complex may be involved in various postinspiratory behaviors such as swallowing and vocalization (Anderson et al., 2016).”

Lines 66-72: please, revise the writing and punctuation and rephrase.

We have removed this text to shorted the introduction and make it more focused.

Line 102: remove semicolon.

This has been removed.

Line 121: please, remove (Figure 1C). You will cite it in the result section.

This has been removed.

Results:In general, writing should be revised to make it less repetitive.

Thank you for your comment. We have revised the Results section, and have removed data in text that is not pertinent to the overall conclusion of this manuscript and have referred the readers to the appropriate supplemental figure and table where all of these parameters are listed. We believe that PiCo neurons not only contain ChATcre/Vglut2FlpO but also ChATcre and Vglut2cre only neurons so it is important to separate out these findings based on the their genetic line. We have however reorganized the subheadings to eliminate the repetitiveness.

Line 127: PiCo region

This has been added.

Lines 172-178: Please, revise the punctuation of the sentence (…mice. Indicating that..)

This has been revised to say: “While there is a moderate degree of correlation in the Vglut2cre:Ai32 (r = 0.33, p< 0.0001, slope = 0.69) mice. This suggests that triggering a swallow in the ChATcre:Ai32 and ChATcre:Vglut2FlpO:ChR2 mice has a stronger effect on resetting the respiratory rhythm than stimulating glutamatergic neurons in Vglut2cre:Ai32 mice.”

Lines 179-180: "…stimulated by ChATcre:Ai32 (r = 0.45, p< 0.0001, slope = 0.35),…" did you mean "stimulated in ChATcre:Ai32 mice"?

Yes, this has been corrected.

Line 182: again why "…mice. Indicating…?

This has been revised to say: “…mice. This suggests that triggering a swallow has a stronger effect on resetting the respiratory rhythm than activating non-swallows in all of the genetic mouse types, under our anesthetic conditions. See discussion for more considerations.

Lines 206-245: Please, avoid the continuous use of parentheses. E.g. instead of.…evoked by water (290{plus minus} 125ms) (199 {plus minus} 125ms) (paired t-test: p = 0.02, t = 3.05, df = 8), it is better, in my opinion:.…evoked by water (290 {plus minus} 125 ms vs. 199 {plus minus} 125 ms; paired t-test: p = 0.02, t = 3.05, df = 8).

This has been changed throughout the text.

Again e.g.: There is a significant decrease in XII (from 297{plus minus} 129ms to 212 {plus minus} 127ms; paired t-test: p = 0.03, t = 2.65, df = 7) and laryngeal complex duration (from 297 {plus minus} 78ms to 163 {plus minus} 63ms; paired t-test: p = 0.009, t = 4.19, df = 5).

This has been changed throughout the text.

Lines 215-216: "max" of what?

Thank you for pointing out that we did not describe this. We have added the following: “Percent of max is measured as a % of the maximum baseline (water swallow) amplitude, see methods.” We have also added this into the methods section. “All durations are reported in milliseconds (ms) and all amplitudes are reported as a “% of max” calculated as the % of the maximum baseline (water swallow) amplitude.”

Line 221: Swallows triggered by stimulation of Vglut2cre:Ai32 neurons…

This has been added.

Lines 277-283: The sentences are unclear. Please, rephrase and avoid the use of "ipsilateral" and "contralateral".

This has been revised.

Reference list:The two references Huff et al. 2022a and 2022b are incomplete (volume, pages).

Thank you for pointing this out. This has been corrected.

Discussion:This section is well-written and detailed, but it should be divided into subheadings to improve clarity and make the writing flow smoothly.

We have added the following subsections into the discussion:

Characteristics of PiCo

Postinspiratory airway protective behaviors

Interactions between PiCo and other brainstem areas

PiCo and the swallow pattern generator

‘All-or-None’ behavior

Schluckatmung

Limitations

Conclusion

A section devoted to methodological considerations, including some limitations of the study, should be introduced.

This has been added.

Attention should be paid to the editing of the references (e.g. line 350).

Thank you for pointing this out. We have fixed the discrepancies within our reference and have edited the in text references to be uniform.

Methods:Line 494: Please, add the ID number of the approved ethic Protocol. The method of euthanasia needs to be explicitly described.

The IACUC protocol number has been added. The method of euthanasia has been added at the end of the in vivo experiments section. It now states: “At the end of the experiment mice are euthanized by an overdose of anesthetic followed by rapid decapitation or trans-cardial perfusion (see histology section below)….. At the end of experiments, animals were deeply anesthetized with 5% isoflurane in 100% oxygen and perfused through the ascending aorta with 20 ml of phosphate buffered saline (PB; pH 7.4) followed by 4% phosphate-buffered (0.1 M; pH 7.4; 20 ml) paraformaldehyde (Electron Microscopy Sciences, Fort Washington, PA).”

Line 512: Did you evaluate the spread of the 150 nL injectate?

We did evaluate the spreading based on counting the transfected neurons. This area had a range of 400 µm rostral-caudal and 650 µm medial-lateral, matching with the area defined as PiCo. Unfortunately, we were not able to measure the spread of the viral solution injected.

Line 527: 1.5 mg/kg

This has been corrected.

Line 531: Please, add the supplemental dose of urethane necessary to maintain anaesthesia.

This has been added. It now states: “A supplemental dose of 0.1mL of Urethane was given to maintain adequate anesthetic depth.”

Line 547:.…the XII and X nerves were isolated unilaterally, cut distally, and their activities were recorded from…

This has been corrected.

Line 554: delete "and"

This has been removed.

Line 557: delete semicolon

This has been removed.

Line 564: delete "postinspiratory complex"

This has been removed.

Lines 566-572: the entire section should be rewritten to improve clarity (no numbered list).

This has been revised. The text now states: “Multiple bipolar EMGs, using 0.002” and 0.003” coated stainless steel wires (A-M Systems, Sequim, WA, USA, part no.790600 and 79100 respectively), simultaneously recorded activity from several swallow and respiratory-related muscle sites. According to the techniques of Basmajian and Stecko (Basmajian and Stecko, 1962), the electrodes were placed using hypodermic needles 30G (part no 305106, BD Precision Glide, Franklin Lakes, NJ, USA) in the submental complex, which consists of the geniohyoid, mylohyoid and digastric muscles, to determine swallow activity. The laryngeal complex, consisting of the posterior cricoarytenoid, lateral, transverse and oblique arytenoid, cricothyroid and thyroarytenoid muscles, to determine laryngeal activity during swallow, as well as postinspiratory activity. The costal diaphragm, used to measure the multifunctional activity for both inspiration, as well as Schluckatmung, a less common diaphragmatic activation during swallow activity (Figure S6).”

Line 582:.… in response to laser stimulation applied to the PiCo region:….

This has been corrected.

Lines 587-588:.…and absence of the diaphragm electromyographic activity.

This has been corrected.

Line 593:.… to the termination of the submental complex electromyographic activity.

This has been corrected.

Line 599: did you mean the onset of the subsequent breathing?

Yes, thank you. This has been corrected.

Lines 657-659: Please, clarify here how did you chose the reference section.

The following has been added: “Section alignment between specimens was done relative to a reference section. The rostral segment of PiCo was identified by the last section with the caudal end of the facial motor neurons and the first section with the rostral portion of the inferior olives. To distinguish PiCo in each section, we used the nucleus ambiguus (Amb), the inferior olives (IO) and the ventral spinocerebellar tract (vsc) as the main anatomic structures. The section that contains the rostral portion of Amb (more densely packed, i.e. cAmb) is the section that contains the rostral portion of PiCo, in a caudal direction, the compacta portion of Amb turns into a semi-compacta portion (scAmb), being aligned as the zero point in the rostral-caudal graphs. Further caudal, the scAmb turns in the non-compacta portion of Amb (Akins et al., 2017; Baertsch et al., 2018; Kottick et al., 2017; Vann et al., 2018), characterizing the caudal edge of PiCo. PiCo was also anatomically characterized by immunohistological labeling, revealing ChAT-positive neurons located dorsomedial to c-scAmb and caudal to the facial nucleus as previously described (Ain Summan Toor et al., 2019; Anderson et al., 2016)”.

Figures:Figure 1. Please, revise the following sentence since it is confounding and there are discrepancies with the plot: (Bii) There is no change in probability of stimulating laryngeal activation between ChATcre:Ai32 and ChATcre:Vglut2FlpO:Ai32 mice, however there is a significant difference between Vglut2cre:Ai32 and * and ChATcre:Vglut2FlpO:Ai32 mice at 70% (p = 0.01) and 90% (p = 0.003) of the respiratory cycle and Vglut2cre:Ai32 and # ChATcre:Ai32 mice at 90% (p = 0.006) of the respiratory cycle.

This has been revised. It now states: “(Bii) There is no change in the probability of stimulating laryngeal activation between ChATcre:Ai32 and ChATcre:Vglut2FlpO:ChR2 mice. However, there is a significant difference in the probability between Vglut2cre:Ai32 (purple) and ChATcre:Vglut2FlpO:ChR2 (gold) mice at ^**^70% (p = 0.01) and ^**^90% (p = 0.003) of the respiratory cycle and Vglut2cre:Ai32 and ChATcre:Ai32 (green) mice at ^##^90% (p = 0.006) of the respiratory cycle.”

Figure 5. It is much better to use as reference point the caudal portion of the VII nucleus (see Anderson et al. 2016). So, please change the coordinates accordingly. It is very hard to detect and discriminate the transition point of the compact and semi-compact portions of the nucleus ambiguus, especially with fluorescence images. How did you do that? In addition, it should be useful to see lower magnification sections along with a histological control stained, for example, with cresyl violet to better appreciate the location and anatomical landmarks of the investigated region.

We apologize for not including the detailed anatomical description of PiCo. We used the cAmb, scAmb, VII, the IO and the vsc as the main anatomic structures. A very detailed description of PiCo has been included in the histology section of the manuscript.

We thank the reviewer for the suggestion and lower magnification pictures have been added to Figure 5, representing the transfected neurons and ChAT staining. However, the cresyl violet staining do not show clearly anatomical landmarks in the area analyzed and we believe this technique does not add to our data.

Figure 6: Please, remove ipsilateral and contralateral.

This has been removed.

Figure S4: See comments on Figure 5 on the reference point. In addition, if in panel B3 and D3 did you mean the total number of neurons instead of the average number? In panel C5, are you sure that at the level of VII nucleus the nucleus ambiguous (pars compacta) is still present (cfr. Franklin and Paxinos atlas)?

Sorry for the confusion. No, we meant the average number of ChAT+ neurons found in 4 animals.

Panel C5 shows exactly the transition between the caudal edge of the facial motor neurons and the first section with the rostral edge of Amb (more densely packed, i.e. cAmb).

Reviewer #6 (Recommendations for the authors):I have reviewed the previous and current submission, the other Reviewers' comments, and the responses of the corresponding author. My comments are brief as the other Reviewers have noted the caveats, many of which I agree with.This study is a tour de force combining technically-demanding approaches to obtain EMG recordings from most of the relevant muscles and nerves (submental complex, laryngeal complex, diaphrgam; vagus and hypoglossal nerves), coupled with elegant intersectional genetic approaches to specifically target for activation cholinergic/glutamatergic neurons that comprise PiCo. This strategy avoids nearby glutamatergic neurons in the reticular formation and cholinergic motoneurons in the NA.The results are clear and striking in that activation of PiCo is sufficient to elicit swallow and laryngeal activation, with the precise timing of each dependent on the phase of the respiratory cycle (swallowing in early PI, laryngeal activation in later I). Moreover, the sequence of events resulting from PiCo activation appear qualitatively similar to those events occuring during a more "physiological" swallow elicited by water administration into the oral cavity. The authors have included the necessary controls, as requested by the Reviewers.In addition, the authors have further refined the anatomical boundaries of Pico using their intersectional transgenic strategy, showing that Glut/Cholinergic neurons that comprise PiCo are medial to the cholinergic neurons of the NA.The authors should be congratulated for performing such a rigorous, technically-demanding set of experiments to show a critical function for PiCo in vivo in swallowing, laryngeal activation, and their coordination with inspiration. Their findings clearly show that PiCo is sufficient to elicit swallowing and reset the respiratory rhythm. They have expanded our knowledge regarding PiCo in these functions and have set the stage for future studies that will further delineate the mechanisms by which PiCo drives these behaviors. No study is perfect, this one included. There are obviously caveats that the other Reviewers have raised, but that have been acknowledged by the authors and, in my opinion, have been discussed sufficiently in the paper. An important extension of these findings is to demonstrate that PiCo is necessary for swallowing/larygneal function (and coordination with breathing) in a more natural, physiological setting (i.e. normal swallow of a bolus of water/food etc). What are the ramifications of optogenetic silencing of PiCo on normal swallowing?

We agree that in this manuscript we have not discussed if PiCo is necessary for natural or physiological response. We have ongoing preliminary data in a disease model (Leigh Syndrome) in which increased inflammation and cell death of PiCo neurons results in swallow-breathing discoordination with water swallows occurring during inspiration instead of postinspiration. When this disease model is treated and the neuroinflammation is eliminated swallow-breathing coordination is restored and swallows occur during postinspiration. With this information we believe PiCo is active during the natural physiologic setting and is responsible for coordinating the proper timing for swallow to occur during postinspiration. We have further preliminary data of using DREADD techniques to silence NTS and find that when NTS is inactivated PiCo cannot trigger swallows. Since all of this work is preliminary it is not appropriate for this manuscript.

To address optogenetically silencing PiCo: While there is a Cre/FlpO AAV for archaerhodopsin we have found the archaerhodopsin to be technically challenging and not sufficient to silence neurons. We used e.g. the archaerhodopsin with Dbx1 neurons in the Huff et al. 2022 manuscript and found this to be the case. We believe a better technique is to use an inhibitory DREADD approach and there is not one currently for cre/FlpO. The Ramirez lab is working on a new AAV approach to knock out PiCo specific neurons however the technology is very novel, and based on a collaboration with another laboratory at the UW and not ready for publication.

In addition, as mentioned by other reviewers, the precise neural circuitry through which PiCo acts, and how it is integrated with the nTS and other regions needs to be more clearly defined. It seems as though the authors are working on these questions in a separate study.

Thank you. The authors agree that looking at the connections between PiCo and NTS is the next step. We have some preliminary data and are in the process of writing a grant to investigate this very question (see above). We have added the following to the manuscript:” These differences are likely due to the fact that the PiCo-triggered swallows are missing the behavioral context of water-evoked swallows and do not activate the sensory component of the SPG to the same extent as the water-evoked swallows. There are known projections from NTS to PiCo (Oliveira et al., 2021) and connections between PiCo and preBötC (Anderson et al., 2016) giving reason to suggest PiCo acts as an interface between the SPG and preBötC.” We originally used the word “gate” to better describe the role of PiCo, but previous reviewers literally hated this description. Thus we opted to delete this word from the manuscript.